# Learning to Adapt Frozen CLIP for Few-Shot Test-Time Domain Adaptation

**Zhixiang Chi**[1], **Li Gu**[2], **Huan Liu**[3], **Ziqiang Wang**[2], **Yanan Wu**[2†],
**Yang Wang**[2†], **Konstantinos N Plataniotis**[1†]

[1] University of Toronto, [2] Concordia University, [3] McMaster University

✉ zhixiang.chi@mail.utoronto.ca

## Abstract

Few-shot Test-Time Domain Adaptation focuses on adapting a model at test time to a specific domain using only a few unlabeled examples, addressing domain shift. Prior methods leverage CLIP's strong out-of-distribution (OOD) abilities by generating domain-specific prompts to guide its generalized, frozen features. However, since downstream datasets are not explicitly seen by CLIP, solely depending on the feature space knowledge is constrained by CLIP's prior knowledge. Notably, when using a less robust backbone like ViT-B/16, performance significantly drops on challenging real-world benchmarks. Departing from the state-of-the-art of inheriting the intrinsic OOD capability of CLIP, this work introduces learning directly on the input space to complement the dataset-specific knowledge for frozen CLIP. Specifically, an independent side branch is attached in parallel with CLIP and enforced to learn exclusive knowledge via revert attention. To better capture the dataset-specific label semantics for downstream adaptation, we propose to enhance the inter-dispersion among text features via greedy text ensemble and refinement. The text and visual features are then progressively fused in a domain-aware manner by a generated domain prompt to adapt toward a specific domain. Extensive experiments show our method's superiority on 5 large-scale benchmarks (WILDS and DomainNet), notably improving over smaller networks like ViT-B/16 with gains of **+5.1** in F1 for iWildCam and **+3.1%** in WC Acc for FMoW. Our Code: L2C

## 1 Introduction

Deep models excel when test and training data distributions align, but real-world scenarios often involve domain shifts (Gulrajani & Lopez-Paz, 2020; Taori et al., 2020; Su et al., 2024b; Yang et al., 2024a; 2023), leading to performance drop. Few-shot Test-Time Domain Adaptation (FSTT-DA) (Chi et al., 2024; Zhong et al., 2022) addresses this by introducing a test-time learning phase to adapt generic models to unseen target domains using a few unlabeled samples. It faces several challenges: *i)* limited domain-specific information due to a few unlabeled data (unseen target domains), *ii) one-time adaptation* for each target domain, *iii)* strict source-free environment during test-time on unseen target domains, and *iv)* handling diverse target domains with varying complexities and domain shifts.

Therefore, developing an adaptive learning system using source domain data is crucial (Ahmed et al., 2021), as it embodies dataset-specific knowledge—including labels, semantics, and domains. MetaDMoE (Zhong et al., 2022) adapts to unseen target domains by querying relevant knowledge from source expert models and then updating an adaptive student model through knowledge distillation (Ye et al., 2024; Yang & Ye, 2024; Hamidi et al., 2024). MABN (Wu et al., 2024b) learns source distributions during offline training and pinpoints domain-specific parameters for updates during test-time. However, both MetaDMoE and MABN involve model fine-tuning, which can compromise the inherent OOD generalization of vision foundation models like CLIP (Wortsman et al., 2022b).

VDPG (Chi et al., 2024) leverages CLIP's inherent OOD capabilities and robustness (Zhang et al., 2023) by operating solely on its visual features. It compacts source domain knowledge into a learnable knowledge bank. A generator then creates domain-specific prompts from this bank, conditioned on

---

[†]Corresponding authors.

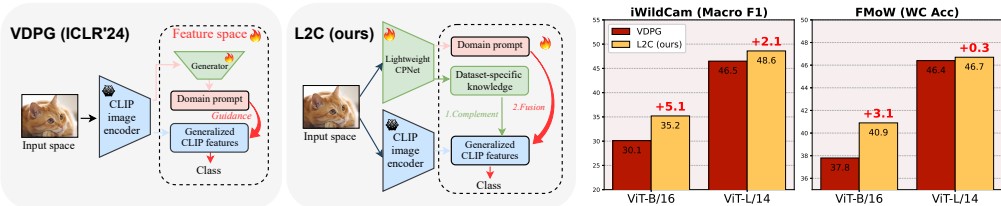

(a) High-level comparison.  (b) Comparison on ViT-B/16 and ViT-L/14.

Figure 1: VDPG preserves CLIP's OOD capabilities by operating in the feature space and relying on pretrained CLIP. However, with a weaker backbone like ViT-B/16, performance drops on benchmarks like iWildCam and FMoW. Learning directly from the image space complements CLIP, improving ViT-B/16 as shown in (b).

the features of unlabeled data, to steer the frozen CLIP features toward the target domain. While effective in generating diverse prompts across domains, VDPG has notable drawbacks: its ability to produce domain-specific prompts and utilize source knowledge is limited by CLIP's general, non-dataset-specific knowledge. As shown in Fig. 1b, with a much less robust backbone, ViT-B/16, VDPG suffers significant performance drop. Moreover, VDPG overlooks class semantic cues (Yoon et al., 2024), which are vital for downstream datasets but not addressed by its vision-only encoder.

In this work, we adopt VDPG's black-box approach to retain CLIP's OOD capabilities and aim to improve it by adapting both frozen image and text features. On the image side, we attach a parallel module named **CPNet** to learn directly from the input space and Com**P**lement the frozen visual feature of CLIP at its output. It contradicts the methods that involve feature interactions among intermediate layers (Yin et al., 2023; Xu et al., 2023; Chi et al., 2018), and allows CPNet to remain lightweight and applicable for black-box settings. CPNet is encouraged via revert attention to focus on learning only the necessary dataset-specific information, both semantic and domain, that may be absent from CLIP's generalized knowledge. Fig. 1a demonstrates high-level comparison with VDPG.

Noting the significant diversity in image features across domains, even for the same group of classes, we deduce that text features must also adapt accordingly. To benefit downstream adaptation, we aim to enhance the discrimination among text features (termed inter-dispersion) (Cho et al., 2023a), reducing domain bias prior to adaptation. We propose a greedy text ensemble strategy to select prompt templates that improve this discrimination, combined with a lightweight refinement module that uses an inter-dispersion loss to further enhance class differentiation. Importantly, because the greedy ensemble is executed as a pre-processing step, the CLIP text encoder can be *discarded when training starts*, minimizing its impact on overall training costs (less than 0.01% of total cost).

To adapt the complemented visual and enhanced text features towards the unseen target domain, we take advantage of CPNet which extracts the unique domain knowledge that CLIP may exclude. Specifically, we reshape the batched unlabeled data so that the inter-attention is computed among the batch instances. Within the same domain, the domain information is typically consistent across data instances (Zhong et al., 2022; Chi et al., 2024). It allows us to treat the propagated batch information as domain-specific knowledge. We then integrate it with a learnable domain cache to form a domain-specific prompt. This prompt guides the fusion of text and image features, enhancing the coherence of domain-specific outputs, and thus adapting to a particular target domain.

We name our framework as **L**earning **to C**omplement (**L2C**) and our contributions are: 1) We propose a parallel CPNet to learn dataset-specific knowledge to complement the generalized frozen CLIP visual feature; 2) We propose effortless greedy ensemble and lightweight refinement to enhance the class-wise inter-dispersion for text features to benefit adaptation; 3) We improve the domain knowledge extraction process to adapt both text and visual features in a domain-aware manner; 4) We evaluate L2C on 5 benchmarks, especially on challenging real-world WILDS dataset with smaller backbones (i.e., **+5.1** in F1 for iWildCam and **+3.1%** in WC Acc for FMoW with ViT-B/16).

## 2 RELATED WORK

Distribution shifts often degrade the learning-based methods (Zhang et al., 2021a). To address this, Domain Generalization (DG) (Zhou et al., 2020; Lv et al., 2022) and Unsupervised Domain Adaptation (UDA) (Zhang, 2021; Peng et al., 2019; Pei et al., 2018) have been explored. DG extracts

domain-invariant features for multiple domains (Li et al., 2018; Long et al., 2018), but a single model often falls short. UDA adapts source knowledge to unlabeled target data through extensive target-specific training, but its scale and resource demands limit practicality. PØDA (Fahes et al., 2023) and ULDA (Yang et al., 2024b) achieve zero-shot adaptation by leveraging natural language descriptions of target domains without accessing data. In contrast, FSTT-DA uses domain cues from target domain images, making it suitable for scenarios where descriptions or labels are unavailable.

Test-time adaptation (TTA) (Su et al., 2022; 2024c;a) is an emerging learning paradigm that incorporates an additional learning phase at test time before inference, to mitigate distribution shifts. This phase often utilizes unsupervised objectives like entropy minimization (Wang et al., 2021; Niu et al., 2022; Zhang et al., 2022a; Gong et al., 2022; Zhao et al., 2023; Wang et al., 2024), teacher-student self-training (Yuan et al., 2023; Marsden et al., 2022; 2023; Liu et al., 2022a; Zhang et al., 2024), auxiliary tasks (Sun et al., 2020; Liu et al., 2023; Chi et al., 2021; Liu et al., 2022b), and contrastive learning (Chen et al., 2022a; Wu et al., 2023; Liang et al., 2022) for supervision. Although effective, these approaches often require model fine-tuning or a complex design of learnable parameters. It challenges their scalability and intrinsic OOD capabilities in larger foundation models (Wortsman et al., 2022a). Recent developments include the use of vision prompts (Han et al., 2023), which adapt by modifying only a minimal number of parameters to leverage the existing knowledge within large models (Zhang et al., 2021b; Gan et al., 2023). HybridPrompt (Wu et al., 2024a) and ProD (Wu et al., 2024a) introduce prompt-based algorithms to extract domain-specific knowledge to address domain shifts in cross-domain few-shot learning (CD-FSL) where labelled support set is avaibale (Guo et al., 2020; Wang & Deng, 2021; Fu et al., 2023). However, these prompts are inserted into various layers and require access to the weights of the base model. Therefore, they incur additional computational costs and pose challenges in scenarios where privacy concerns or proprietary models limit flexibility (An et al., 2022). Our work introduces a practical, gradient-free adaptation method, enabling model deployment in black-box environments.

**Few-shot test-time domain adaptation (FSTT-DA).** FSTT-DA utilizes a few unlabeled data samples for domain adaptation, providing a practical edge over instance-level methods. MetaDMoE (Zhong et al., 2022) separately trains a pool of domain-specific experts, a process that creates boundaries in knowledge transfer among source domains. MABN (Wu et al., 2024b) focuses on identifying and updating domain-specific parameters via a self-supervised auxiliary branch. It makes its effectiveness dependent on the auxiliary task. VDPG (Chi et al., 2024) harnesses the inherent OOD generalization capabilities of VFMs (Zhang et al., 2023) to create a domain prompt generator that aligns VFM features to specific domains, yet it is limited by a lack of dataset-specific knowledge. In contrast, our method directly learns from the input space, effectively integrating dataset-specific knowledge with the robust OOD capabilities of foundation models.

**Efficient tuning with side network.** Recent trends favor employing a smaller, parallel side network over inserting learnable parameters into the main backbone. This approach has proven effective in dense prediction (Chen et al., 2022b; Xu et al., 2023) and recognition tasks (Fu et al., 2024; Wang et al., 2023; Sung et al., 2022). However, these methods typically require accessing or modifying the main backbone's intermediate features for efficient adaptation. Our proposed framework diverges by integrating a revert attention mechanism that learns dataset-specific knowledge, aiming to enhance the output of pre-trained foundation models without intervening in their internal processes.

## 3 PRELIMINARIES

**Problem setting.** In this study, we address Few-Shot Test-time Domain Adaptation (FSTT-DA) (Zhong et al., 2022; Wu et al., 2024b; Chi et al., 2024). In this setting, a model is trained on $N$ labeled source domains $\mathcal{D}_s = \{\mathcal{D}_s^n = (x_s, y_s)^n\}_{n=1}^N$, and then is tested on $M$ target domains with only input images: $\mathcal{D}_t = \{\mathcal{D}_t^m = (x_t)^m\}_{m=1}^M$. We assume distribution shifts occur between any source and target domain pairs and they share the same label space $\mathcal{Y}_s = \mathcal{Y}_t$. At test-time, for each target domain $\mathcal{D}_t^m$, a few-shot of unlabeled data $\mathbf{x}$ is used to adapt the model which is used for inference on $\mathcal{D}_t^m$. This adaptation stage is source-free, as source data are not used post-training. Appendix B depicts the setting of FSTT-DA.

**Motivations.** Foundation models like CLIP, trained on web-scale datasets (Oquab et al., 2023; Radford et al., 2021), have markedly improved downstream tasks (Cho et al., 2023b; Goyal et al., 2023). However, it remains a significant challenge to adapt these models to unseen domains using

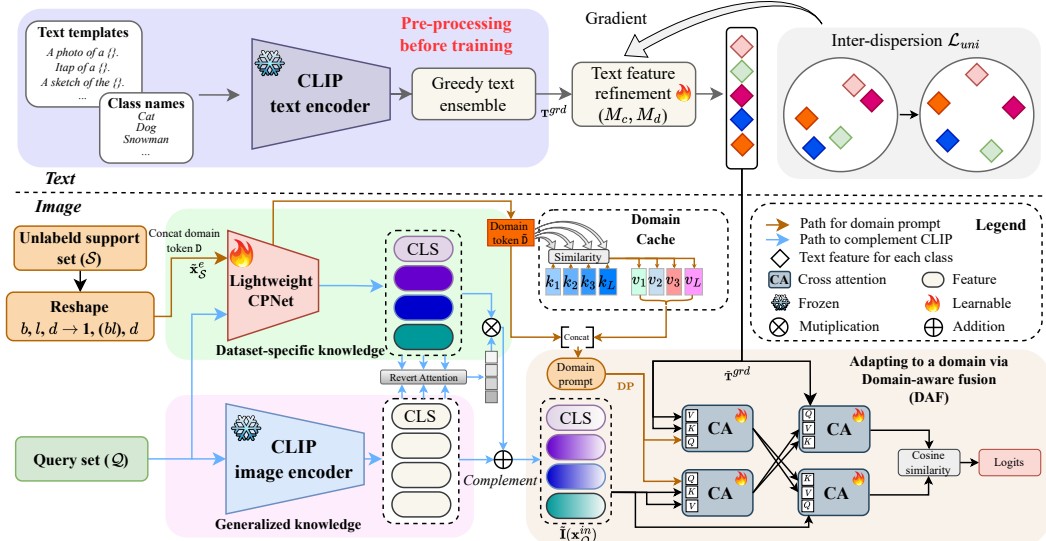

Figure 2: Training process of L2C on source domains. **(Top)** For a dataset, our greedy strategy selects text prompts with larger inter-dispersion which will be refined subsequently ($\tilde{\mathbf{T}}^{gre}$). **(Bottom)** CPNet is proposed in parallel with CLIP image encoder to learn dataset-specific knowledge to complement the generalized knowledge in CLIP. To adapt to a domain, a few unlabeled data samples (support set $\mathcal{S}$) are used to first generate a domain prompt **DP** via CPNet and domain cache. **DP** is then used to adapt all the data (query set $\mathcal{Q}$ with image feature: $\tilde{\mathbf{I}}(\mathbf{x}_\mathcal{Q}^{in})$ and text feature: $\tilde{\mathbf{T}}^{gre}$) in that domain via domain-aware fusion (DAF).

minimal unlabeled data as in FSTT-DA. A key approach (VDPG) has been proposed to harness their inherent OOD generalization capabilities (Chi et al., 2024). This involves using domain-specific prompts based on a few data features to adapt CLIP's broad features to particular domains. Nevertheless, CLIP has not specifically seen the downstream datasets. The method of deriving domain-specific knowledge strictly from a generalized feature space has inherent limitations. Consequently, VDPG's reliance on CLIP's pre-trained knowledge restricts its performance. As shown in Fig. 1b and Table 1, using a weaker model like ViT-B/16 yields poor results on the challenging WILDS benchmarks. These shortcomings have motivated us to develop an efficient framework that learns directly from the input space. Our approach not only taps into *dataset-specific knowledge including semantics and distribution/domain cues* to complement generalized CLIP features, but also leverages text features to enrich label semantics, thus significantly enhancing adaptation capability.

## 4 METHOD

**Overview.** We aim to adapt both image and text features to unseen domains, as illustrated in Fig. 2. In Sec.4.1, we introduce CPNet to acquire dataset-specific knowledge, complementing CLIP's visual features. Sec.4.2 covers our greedy ensemble approach and lightweight refinement for enhancing text features. In Sec.4.3, we first demonstrate the generation of domain-specific prompts and then adapt the features using domain-aware fusion. Sec.4.4 outlines the training and inference process.

### 4.1 LEARNING DATASET-SPECIFIC VISUAL KNOWLEDGE TO COMPLEMENT CLIP

**Parallel CPNet.** Freezing CLIP is effective in retaining its OOD capability (Wortsman et al., 2022b). We propose an independent CPNet in parallel with the CLIP image encoder to learn dataset-specific knowledge to complement CLIP. We use boldface $\mathbf{x}$ to refer to a batch of images and use $x$ to indicate one image. Given an image $x \in \mathbb{R}^{1 \times H \times W \times C}$, it is first split into $l$ patches and encoded into embeddings with dimension $d$. A class token $\texttt{[CLS]}$ is pre-pended to form the input $(in)$ tokens as $x^{in} \in \mathbb{R}^{1 \times (l+1) \times d}$. Let $\mathbf{I}$ represent the CLIP image encoder, we denote its output as $\mathbf{I}(x^{in}) \in \mathbb{R}^{1 \times (l+1) \times d}$. We impose minimal architectural constraints on CPNet but only match its output dimension with that of the CLIP encoder, thus we express CPNet as $\mathbf{CP}(x^{in}) \in \mathbb{R}^{1 \times (l+1) \times d}$.

Given that CLIP has mastered extensive generalized knowledge, it is strategically beneficial for CPNet to only acquire *necessary dataset-specific* semantic and domain information not encompassed by CLIP. Consequently, we introduce a parameter-free Revert Attention (RT) mechanism (Chen et al., 2018) to specifically target the learning of CPNet. We employ the Scaled Dot-Product Attention method (Vaswani et al., 2017) to calculate the attention between their outputs and then compute its complement with respect to $\mathbf{1}$. The resulting reverted attention map $\mathbf{A}$ is reapplied to $\mathbf{CP}(x^{in})$ using a dot product:

$$\mathbf{CP}^{RT}(x^{in}) = \mathbf{A} \cdot \mathbf{CP}(x^{in}), \quad \text{where } \mathbf{A} = \mathbf{1} - \text{softmax}(\mathbf{CP}(x^{in}) \cdot \mathbf{I}(x^{in})), \tag{1}$$

where $\cdot$ represents the dot-product. This approach ensures that CPNet is focused solely on learning information distinctive from CLIP, rendering efficiency and compactness (e.g., requiring only 3 transformer blocks to complement 12-layer ViT-B/16 on the DomainNet dataset). The dataset-specific information is added back to complement the CLIP visual feature: $\tilde{\mathbf{I}}(x^{in}) = \mathbf{I}(x^{in}) + \mathbf{CP}^{RT}(x^{in})$.

### 4.2 ENHANCING THE LEARNING ON DATASET-SPECIFIC LABEL SEMANTICS

For classification with CLIP, text features act as class prototypes, generated by pairing class names with a template (e.g., `A photo of a [CLASS]`). We freeze CLIP to leverage its OOD generalization, so the same set of text features is shared across domains. In Fig. 3, we calculate the average difference in image embeddings for two selected class pairs across 6 domains in DomainNet. The significant semantic variation, even within the same class pairs, highlights the need for domain-specific text features. Since CLIP relies on the *unified* space of text and image, we focus on adapting text features alongside image features.

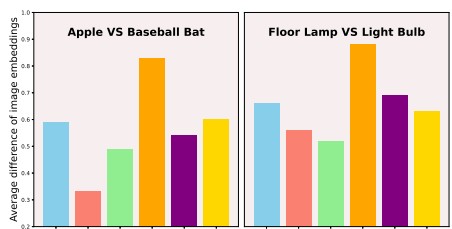

Figure 3: Image embedding differences are calculated as 1 minus cosine similarity for the displayed classes and domains, highlighting that semantic variations differ notably across domains, which requires customized text features for each domain.

To facilitate the adaptation process, we propose reducing domain biases in the text features by increasing their discrimination (inter-dispersion). This ensures that text features remain neutral across all domains before adaptation. To achieve this, we introduce a greedy text ensemble strategy as a preprocessing step, selecting text templates that enhance inter-dispersion. Let $\mathbf{T}(\mathbf{P_C})$ represent the text features from a prompt template P and class labels $\mathbf{C}$ using text encoder $\mathbf{T}$. The inter-dispersion of text features is quantified by their *uniformity* in a hypersphere (Cho et al., 2023a; Wang & Isola, 2020), described as:

$$\mathcal{L}_{uni}(\mathbf{T}(\mathbf{P_C})) = \sum_{i,j \in |\mathbf{C}|, i \neq j} \exp(-t \|\mathbf{T}_i(\mathbf{P_C}) - \mathbf{T}_j(\mathbf{P_C})\|_2^2), \tag{2}$$

where $i, j$ represent $i^{th}$ or $j^{th}$ class, $t = 2$ by default. Assuming $P$ prompt templates, we sort them by increasing $\mathcal{L}_{uni}$ values: $\{\mathbf{T}(\mathbf{P_C^p})\}_{p=1}^P$. We begin our ensemble with the most uniform embedding, $\mathbf{T}(\mathbf{P_C^1})$, and incrementally add others. The $p^{th}$ prompt is retained in the ensemble list $\mathbf{E}$ if it reduces the overall uniformity metric:

$$\mathcal{L}_{uni}(Ave([\mathbf{E}, \mathbf{T}(\mathbf{P_C^p})])) < \mathcal{L}_{uni}(Ave(\mathbf{E})). \tag{3}$$

where *Ave* represents averaging ensemble. After selecting prompts greedily, we ensemble $\mathbf{E}$ into $\mathbf{T}^{gre} \in \mathbb{R}^{|\mathbf{C}| \times d}$ via averaging, a well-informed initialization that encapsulates CLIP's text knowledge. Pseudocode for the greedy ensemble is provided in Appendix E. In Appendix F.2, we show another alternative to quantify inter-dispersion. At this stage, the text encoder $\mathbf{T}$ can be discarded making the ensemble process effortless, occupying less than 0.01% of the total cost as in Appendix D.2. To further improve the inter-dispersion, we introduce a lightweight module to refine $\mathbf{T}^{gre}$ when training on the source data:

$$\tilde{\mathbf{T}}^{gre} = M_c \mathbf{T}^{gre} M_d + \mathbf{T}^{gre}, \tag{4}$$

where $M_c \in \mathbb{R}^{|\mathbf{C}| \times |\mathbf{C}|}$ and $M_d \in \mathbb{R}^{d \times d}$ adjust along label and feature dimensions, respectively. Additionally, we apply the *uniformity loss* at the output as $\mathcal{L}_{uni}(\tilde{\mathbf{T}}^{gre})$.

### 4.3 ADAPTING TO A DOMAIN VIA DOMAIN-AWARE FUSION

**Domain prompt computation.** We aim to compute a domain-specific prompt to adapt both image ($\tilde{\mathbf{I}}(x^{in})$) and text ($\tilde{\mathbf{T}}^{gre}$) features towards unseen target domains. The source domain knowledge is critical in helping the computation of domain prompt (Chi et al., 2024). We follow the Cache-based learning methods (Zhang et al., 2022b; Zhu et al., 2023) to build a learnable key ($\mathbf{K} \in \mathbb{R}^{L \times d}$) - value ($\mathbf{V} \in \mathbb{R}^{L \times d}$) domain cache to store and query such learned source knowledge, where $L$ is the cache size. Given a batch of $b$ unlabeled images $\mathbf{x}$ from a domain in FSTT-DA, $\mathbf{K}$ is used to compute the similarity between that domain and the source domains, which will be used to query the source knowledge from $\mathbf{V}$.

To process $\mathbf{x}$, we first transform it into an embedding $\mathbf{x}^e \in \mathbb{R}^{b \times l \times d}$. VDPG directly feeds $\mathbf{x}^e$ into a transformer. Since the attention mechanism operates along the $l$ dimension, excluding the batch dimension $b$, this results in separate attention for each image in $\mathbf{x}^e$. This approach is non-intuitive, as domain knowledge should be instance-agnostic. Instead, we propose computing interrelations within the batch (Blattmann et al., 2023) using the dataset-specific CPNet. To achieve this, we reshape $\mathbf{x}^e$ by combining the first two dimensions into $\tilde{\mathbf{x}}^e \in \mathbb{R}^{1 \times (b \times l) \times d}$. This allows the attention mechanism to operate along the $(b \times l)$ dimension, interleaving all the images in $\mathbf{x}$.

Analogous to classification, where a `CLS` token aggregates global information for an image, we prepend a learnable domain token (`D`) so that all information in $\tilde{\mathbf{x}}^e$ is propagated to `D` through attention (Dosovitskiy et al., 2020). The prepended token [`D`, $\tilde{\mathbf{x}}^e$] is fed to $\mathbf{CP}$. We then retrieve $\tilde{\text{D}}$ from $\mathbf{CP}$([`D`, $\tilde{\mathbf{x}}^e$]) to query the source domain information from $\mathbf{K}$-$\mathbf{V}$ cache by computing their similarity as $softmax(\mathbf{K}\tilde{\text{D}}^T) \cdot \mathbf{V}$. The domain prompt ($\mathbf{DP}$) is the concatenation of the queried source knowledge and $\tilde{\text{D}}$ which represents the domain-specific knowledge of that domain:

$$\mathbf{DP} \in \mathbb{R}^{(L+1) \times d} = [\text{softmax}(\mathbf{K}\tilde{\text{D}}^T) \cdot \mathbf{V}, \tilde{\text{D}}]. \tag{5}$$

**Domain-aware fusion.** Once the domain prompt $\mathbf{DP}$ is obtained by Eq. 5 with unlabeled data, we aim to adapt all the data $x^{in}$ in that domain. To this end, we propose cross-attentions among $\mathbf{DP}$, $\tilde{\mathbf{I}}(x^{in})$ and $\tilde{\mathbf{T}}^{gre}$ to progressively fuse them with a domain-aware fusion module: DAF($\mathbf{DP}$, $\tilde{\mathbf{I}}(x^{in})$, $\tilde{\mathbf{T}}^{gre}$). Specifically, we first separately project $\tilde{\mathbf{I}}(x^{in})$ and $\tilde{\mathbf{T}}^{gre}$ into that domain by conditioning on $\mathbf{DP}$ using cross-attention ($\mathbf{CA}$) (Jaegle et al., 2021):

$$\mathbf{DP^T} = \mathbf{CA}(K = \tilde{\mathbf{T}}^{gre}, V = \tilde{\mathbf{T}}^{gre}, Q = \mathbf{DP}),$$
$$\mathbf{DP^I} = \mathbf{CA}(K = \tilde{\mathbf{I}}(x^{in}), V = \tilde{\mathbf{I}}(x^{in}), Q = \mathbf{DP}), \tag{6}$$

Note, that we omit the $QKV$ weight matrices and the FFN layer for simplicity. Now, $\mathbf{DP^T}$ and $\mathbf{DP^I}$ contain their modality information in the same domain, we then cross fuse them into other modality:

$$\mathbf{T}^{dm} = \mathbf{CA}(K = \mathbf{DP^T}, V = \mathbf{DP^T}, Q = \tilde{\mathbf{I}}(x^{in})),$$
$$\mathbf{I}^{dm} = \mathbf{CA}(K = \mathbf{DP^I}, V = \mathbf{DP^I}, Q = \tilde{\mathbf{T}}^{gre}), \tag{7}$$

where $dm$ represents domain. We then obtain their class token $I^{dm}$ from $\mathbf{I}^{dm}$ and its corresponding text feature $T^{dm}$ from $\mathbf{T}^{dm}$ as $[(I_1^{dm}, T_1^{dm}), ..., (I_B^{dm}, T_B^{dm})]$, where $B$ is the batch size. We finally follow the original CLIP loss (Goyal et al., 2023; Radford et al., 2021) on the adapted text and image features as:

$$\mathcal{L}_{clip} = \sum_{i=1}^{B} -\log \frac{\exp\left((I_i^{dm}) \cdot (T_i^{dm})\right)}{\sum_{j=1}^{B} \exp\left((I_i^{dm}) \cdot (T_j^{dm})\right)} + \sum_{i=1}^{B} -\log \frac{\exp\left((I_i^{dm}) \cdot (T_i^{dm})\right)}{\sum_{j=1}^{B} \exp\left((I_j^{dm}) \cdot (T_i^{dm})\right)}. \tag{8}$$

The final loss is defined as: $\mathcal{L}_{total} = \mathcal{L}_{clip} + \lambda \mathcal{L}_{uni}(\tilde{\mathbf{T}}^{gre})$, where $\lambda$ balances two losses.

### 4.4 DOMAIN-CENTRIC LEARNING TO ADAPT

**Training on source domains.** Our ultimate goal is to adapt to unseen target domains using only a few unlabeled data samples. It is essential to align the training objective directly with the evaluation protocol, embodying the system learning to adapt. Therefore, instead of uniformly sampling the data across domains, we follow VDPG to learn at the domain level and mimic the adaptation at test-time as in meta-learning (Finn et al., 2017; Chi et al., 2022; Gu et al., 2022).

---

**Algorithm 1** Domain-centric learning to adapt

---

**Require:** **I/T**: CLIP image/text encoders; $\{P^p\}_{p=1}^P$: $P$ text prompt templates; **C**: $C$ classes with names; $\mathcal{D}_s$: source domains; $\alpha$: learning rate; **CP**: CPNet; **K/V**: K-V domain cache; DAF: domain-aware fusion module; $M_c/M_d$: text refinement;

1: **// Greedy text feature ensemble**
2: $\{\mathbf{T}(P_C^p)\}_{p=1}^P$ ▷ Compute and sort text features for all text prompt templates
3: Obtain $\mathbf{T}^{gre}$ via greedy ensemble using Eq. 3, then discard the text encoder.
4: **// Learning to complement CLIP and adapt to a particular domain**
5: **for** itr=1 to Max_iteration **do**
6: $\quad (\mathbf{x}_S), (\mathbf{x}_Q, \mathbf{y}_Q) \sim \mathcal{D}_s^n$ ▷ Sample a source domain and support and query sets
7: $\quad \bar{\mathbf{x}}_S^e \leftarrow \mathbf{x}_S, \quad \mathbf{x}_Q^{in} \leftarrow \mathbf{x}_Q$ ▷ Form input embeddings
8: $\quad \tilde{\mathbb{D}} \leftarrow \mathbf{CP}([\mathbb{D}, \tilde{\mathbf{x}}_S^e])$ ▷ Aggregate domain information from support set
9: $\quad \mathbf{DP} = [\text{softmax}(\mathbf{K}\tilde{\mathbb{D}}^T) \cdot \mathbf{V}, \tilde{\mathbb{D}}]$ ▷ Form a domain prompt for domain $\mathcal{D}_s^n$
10: $\quad \bar{\mathbf{I}}(\mathbf{x}_Q^{in}) \leftarrow \mathbf{I}(\mathbf{x}_Q^{in}) + \mathbf{CP}^{RT}(\mathbf{x}_Q^{in})$ ▷ Compute complemented visual feature
11: $\quad \tilde{\mathbf{T}}^{gre} \leftarrow M_c\mathbf{T}^{gre}M_d + \mathbf{T}^{gre},$ ▷ Refine ensembled text feature
12: $\quad \mathbf{T}_Q^{dm}, \mathbf{I}_Q^{dm} \leftarrow \text{DAF}(\mathbf{DP}, \bar{\mathbf{I}}(\mathbf{x}_Q^{in}), \tilde{\mathbf{T}}^{gre})$ ▷ Adapt query towards domain $\mathcal{D}_s^n$
13: $\quad (\mathbf{CP}, \mathbf{K/V}, \text{DAF}, M_1/M_2) \leftarrow (\mathbf{CP}, \mathbf{K/V}, \text{DAF}, M_1/M_2) - \alpha\nabla\mathcal{L}_{total}$
14: **end for**

---

Algo. 1 and Fig. 2 show our training scheme. For each dataset, the text process is only executed once to obtain the ensembled text feature (L2-3). The entire text encoder can be discarded before official training. For each iteration, we consider it as an adaptation task on a randomly sampled source domain $\mathcal{D}_s^n$. Two disjoint support set $(\mathbf{x}_S)$ and query set $(\mathbf{x}_Q, \mathbf{y}_Q)$ are sampled. $(\mathbf{x}_S)$ is used to generate the domain prompt (L8-9). Then the complemented visual feature is computed for $\mathbf{x}_Q$ and adapted by the domain prompt (L10-12). The whole system is evaluated by the loss on the query set (L13).

**Adapting to unseen target domain at inference.** After iterations of the adaptation task trained on source domains, L2C is ready to adapt to unseen domains. Algo.2 in Appendix C outlines the inference process, which consists of two phases: 1) For each target domain, a few unlabeled data samples are first drawn, and the domain prompt is obtained. Afterward, the K-V cache can be discarded. 2) The domain prompt is then used to adapt every data sample in that domain. Fig. 8a & 8b in Appendix C demonstrate the two phases.

## 5 EXPERIMENTS

**Datasets and evaluation.** We follow VDPG to evaluate on DomainNet (Peng et al., 2019), which comprises 569K images across 345 classes in 6 domains. We follow the official leave-one-domain-out protocol to train 6 models and report accuracy. We also evaluate on 4 WILDS (Koh et al., 2021) benchmarks, known for their real-world challenges and notably low CLIP zero-shot accuracy (Chi et al., 2024). This includes classification benchmarks such as iWildCam (Beery et al., 2021), Camelyon17 (Bandi et al., 2018), and FMoW (Christie et al., 2018). Although CLIP is primarily designed for classification, we also adapt our framework for regression (PovertyMap (Yeh et al., 2020)), detailed in Appendix F.3. Evaluation metrics include accuracy, Macro F1, worst-case accuracy, Pearson correlation (r), and its worst-case.

**Architecture and training details.** We use official CLIP pre-trained ViT-B/16 and ViT-L/14 as the foundation models. Their feature dimensions ($d$) are 768 and 1024 respectively. Therefore, our CPNet is stacked by regular transformer modules as in ViT with the same feature dimensions. The model is trained for 20 epochs with SGD using cosine decay with initial learning rates of $2.5e^{-3}$ and $1e^{-3}$ for WILDS and DomainNet. $\lambda$ is set to 0.1 to balance the losses. We use 16 images for adaptation at inference. Appendix G&H lists additional hyperparameters and the text prompts.

### 5.1 MAIN RESULTS

**Evaluation on WILDS.** The WILDS benchmarks reveal complex real-world domain shifts, like wild-camera setups, remote sensing, and medical imaging. It is characterized by significant data imbalances at domain and class levels. CLIP demonstrates notably low zero-shot accuracies in these scenarios. However, as Table 1 indicates, our method substantially exceeds previous approaches. It surpasses VDPG with improvements of **2.1 and 5.1 in Macro-F1 for iWildCam**, and enhances **WC Acc by 0.3% and 3.1% for FMoW** with ViT-L/14 and ViT-B/16, respectively. ViT-B/16 shows notably weaker learning capabilities compared to ViT-L/14. Our method, which learns directly from the input space, effectively harnesses domain-specific and data-specific knowledge, thus outperforming VDPG, particularly in models with lower capacities (i.e., ViT-B/16) across diverse WILDS datasets.

**Evaluation on DomainNet.** Table 2 presents the accuracy across various domains and their overall averages. Our approach significantly surpasses VDPG, in **4/6** and **5/6** domains with average accuracy

Table 1: Evaluation on challenging WILDS image testbeds under OOD conditions. It reveals that our method excels in both classification and regression tasks, significantly outperforming SOTA methods. Notably, with a smaller network (ViT-B/16), our method surpasses VDPG due to independently learned data-specific knowledge. (*: results obtained using official code; †: main evaluation metrics in WILDS; ◇: 3/8 channels utilized in PovertyMap as in VDPG.)

| Method | Backbone | iWildCam | | Camelyon17 | FMoW | | PovertyMap◇ (Regression) | |
|---|---|---|---|---|---|---|---|---|
| | | Acc | Macro F1† | Acc† | WC Acc† | Avg Acc | WC Pearson r† | Pearson r |
| ERM | | 71.6 (2.5) | 31.0 (1.3) | 70.3 (6.4) | 32.3 (1.25) | 53.0 (0.55) | 0.45 (0.06) | 0.78 (0.04) |
| CORAL | | 73.3 (4.3) | 32.8 (0.1) | 59.5 (7.7) | 31.7 (1.24) | 50.5 (0.36) | 0.44 (0.06) | 0.78 (0.05) |
| IRM | | 59.8 (3.7) | 15.1 (4.9) | 64.2 (8.1) | 30.0 (1.37) | 50.8 (0.13) | 0.43 (0.07) | 0.77 (0.05) |
| ARM-CML | CNNs | 70.5 (0.6) | 28.6 (0.1) | 84.2 (1.4) | 27.2 (0.38) | 45.7 (0.28) | 0.37 (0.08) | 0.75 (0.04) |
| ARM-BN | | 70.3 (2.4) | 23.7 (2.7) | 87.2 (0.9) | 24.6 (0.04) | 42.0 (0.21) | 0.49 (0.21) | **0.84 (0.05)** |
| Meta-DMoE | | 77.2 (0.3) | 34.0 (0.6) | 91.4 (1.5) | 35.4 (0.58) | 52.5 (0.18) | 0.51 (0.04) | 0.80 (0.03) |
| MABN | | **78.4(0.6)** | **38.3(1.2)** | 92.4(1.9) | **36.6(0.41)** | **53.2(0.52)** | **0.56 (0.05)** | **0.84 (0.04)** |
| Zero-shot (ZS) | ViT-B/16 | 14.9 | 9.7 | 50.1 | 14.5 | 16.3 | 0.27 | 0.58 |
| VDPG* | CLIP | 71.4 (0.2) | 30.1 (0.3) | 93.2 (0.3) | 37.8 (0.5) | 52.7 (0.3) | 0.38 (0.02) | 0.77 (0.02) |
| **L2C (ours)** | | **73.4 (0.4)** | **35.2 (0.3)** | **94.2 (0.2)** | **40.9 (0.4)** | **54.8 (0.1)** | **0.50 (0.02)** | **0.80 (0.03)** |
| Zero-shot (ZS) | | 28.7 | 1.0 | 64.2 | 13.3 | 21.1 | 0.35 | 0.62 |
| FLYP | ViT-L/14 | 72.2 (0.4) | 41.9 (0.3) | - | 46.0 (0.3) | **63.3 (0.4)** | - | - |
| VDPG | CLIP | **78.8 (0.2)** | 46.5 (0.3) | 96.0 (0.4) | 46.4 (0.5) | 61.9 (0.4) | 0.51 (0.03) | 0.83 (0.04) |
| **L2C (ours)** | | 77.3 (0.1) | **48.6 (0.4)** | **96.1 (0.3)** | **46.7 (0.3)** | 61.4 (0.3) | **0.56 (0.02)** | **0.84 (0.03)** |

Table 2: Evaluation on DomainNet. Our method surpasses SOTA, achieving top accuracy in **4/6** and **5/6** domains, with average gains of **+1.4%** and **+2.2%** using ViT-B/16 and ViT-L/14, respectively.

| Method | Backbone | Clip | Info | Paint | Quick | Real | Sketch | Avg. |
|---|---|---|---|---|---|---|---|---|
| ERM | | 58.1 (0.3) | 18.8 (0.3) | 46.7 (0.3) | 12.2 (0.4) | 59.6 (0.1) | 49.8 (0.4) | 40.9 |
| Mixup | | 55.7 (0.3) | 18.5 (0.5) | 44.3 (0.5) | 12.5 (0.4) | 55.8 (0.3) | 48.2 (0.5) | 39.2 |
| CORAL | | 59.2 (0.1) | 19.7 (0.2) | 46.6 (0.3) | 13.4 (0.4) | 59.8 (0.2) | 50.1 (0.6) | 41.5 |
| MTL | CNNs | 57.9 (0.5) | 18.5 (0.4) | 46.0 (0.1) | 12.5 (0.1) | 59.5 (0.3) | 49.2 (0.1) | 40.6 |
| SegNet | | 57.7 (0.3) | 19.0 (0.2) | 45.3 (0.3) | 12.7 (0.5) | 58.1 (0.5) | 48.8 (0.2) | 40.3 |
| ARM | | 49.7 (0.3) | 16.3 (0.5) | 40.9 (1.1) | 9.4 (0.1) | 53.4 (0.4) | 43.5 (0.4) | 35.5 |
| Meta-DMoE | | 63.5 (0.2) | 21.4 (0.3) | 51.3 (0.4) | 14.3 (0.3) | 62.3 (1.0) | 52.4 (0.2) | 44.2 |
| MABN | | 64.2 | 23.6 | 51.5 | 15.2 | 64.6 | 54.1 | 45.5 |
| DoPrompt | ViT-B/16 IMN | 67.6 (0.2) | 24.6 (0.1) | 54.9 (0.1) | 17.5 (0.2) | 69.6 (0.3) | 55.2 (0.5) | 48.3 |
| Zero-shot (ZS) | | 69.9 | 48.2 | 65.4 | 14.5 | 82.3 | 62.5 | 57.1 |
| ERM | ViT-B/16 | 68.0 (0.1) | 22.5 (0.6) | 46.5 (4.2) | 18.5 (0.9) | 58.7 (2.7) | 52.5 (1.2) | 44.4 |
| MIRO | CLIP | 74.9 (0.2) | 37.1 (0.4) | 59.8 (0.6) | **18.7 (1.2)** | 72.2 (0.2) | 61.2 (0.9) | 54.0 |
| VDPG | | **76.3 (0.2)** | 49.3 (0.4) | 67.8 (0.1) | 17.4 (0.2) | 81.5 (0.3) | 66.6 (0.2) | 59.8 |
| **L2C (ours)** | | 75.6 (0.1) | **52.1 (0.1)** | **69.4 (0.1)** | 17.3 (0.2) | **85.5 (0.1)** | **67.1 (0.2)** | **61.2** |
| Zero-shot (ZS) | ViT-L/14 | 78.1 | 54.0 | 71.6 | 21.8 | 86.0 | 71.2 | 63.8 |
| VDPG | CLIP | **82.4** | 54.9 | 73.1 | 22.7 | 85.0 | 73.2 | 65.2 |
| **L2C (ours)** | | 82.3 | **58.7** | **75.2** | **24.0** | **88.6** | **75.4** | **67.4** |

improvements of **+1.4** and **+2.2** using ViT-B/16 and ViT-L/14, respectively. These improvements underscore the benefits of directly learning dataset-specific knowledge. In Appendix F.1, we further compare with prompt-based methods: CoOp (Zhou et al., 2022b), CoCoOp (Zhou et al., 2022a) and side branch-based DTL (Fu et al., 2024)

## 5.2 ABLATION STUDIES

We conduct ablation studies on DomainNet-Info, iWildcam and FMoW using CLIP ViT-B/16 on various components, including CPNet, Revert Attention (RT), text refinement (Text ref.), greedy ensemble (Greedy), uniformity loss ($\mathcal{L}_{uni}$), DAF module and training schemes in Table 3. Note, if DAF is not incorporated, the domain branch is also omitted, which will be discussed in Table 4.

**CPNet and revert attention.** As highlighted in *Index 1 vs. 2* of Table 3, incorporating CPNet alongside a frozen CLIP markedly enhances performance across all datasets. On particularly challenging iWildCam and FMoW, there is exceptionally low zero-shot performance. However, effective learning of dataset-specific knowledge from the input space results in substantial performance gains, specifically, **+13.1 on F1** for iWildCam and **13.8% WC Acc** for FMoW. Additionally, integrating reverted attention directs CPNet to assimilate knowledge overlooked by CLIP, sharpening its focus on essential dataset-specific insights. This strategy leads to further enhancements (*Index 4 vs. 5*).

Table 3: Ablation on various components of our work on DomainNet-Info, iWildCam and FMoW.

| Index | CPNet | Text ref. | Greedy | $\mathcal{L}_{uni}$ | DAF | Training | Info | iWildCam | | FMoW | |
|---|---|---|---|---|---|---|---|---|---|---|---|
| | | | | | | | Acc | Acc | F1 | WC Acc | Acc |
| 1 (ZS) | - | - | - | - | - | - | 48.2 | 14.9 | 9.7 | 14.5 | 16.3 |
| 2 | ✓ | - | - | - | - | ERM | 49.6 | 65.8 | 22.8 | 28.3 | 49.0 |
| 3 | ✓ | ✓ | - | - | - | ERM | 50.3 | 68.7 | 24.0 | 31.4 | 50.9 |
| 4 | ✓ | ✓ | ✓ | - | - | ERM | 51.0 | 69.1 | 26.9 | 33.3 | 51.6 |
| 5 | RT | ✓ | ✓ | - | - | ERM | 51.5 | 71.5 | 32.9 | 36.1 | 53.1 |
| 6 | RT | ✓ | ✓ | ✓ | - | ERM | 51.7 | 72.2 | 33.2 | 36.0 | 53.8 |
| 7 | RT | ✓ | ✓ | ✓ | ✓ | ERM | 51.5 | 71.9 | 32.8 | 36.0 | 52.8 |
| 8 | RT | ✓ | ✓ | ✓ | ✓ | Domain-centric | 52.1 | 73.4 | 35.2 | 40.9 | 54.8 |

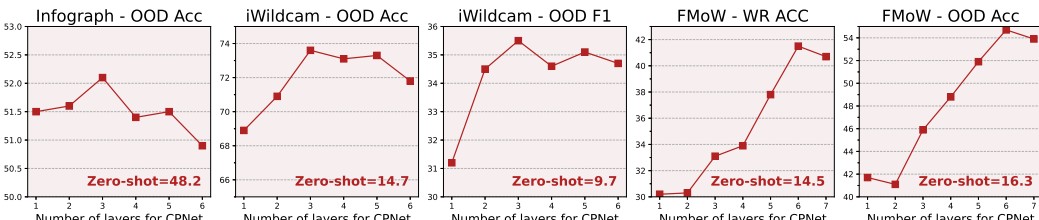

Figure 4: Analysis on a different number of transformer layers of CPNet.

**Text refinement and greedy ensemble.** Text features from frozen CLIP serve as initialization and refining them as per Eq. 4 proves beneficial for subsequent adaptation (*Index 2 VS. 3*). Since text features act as class prototypes, enhancing class discrimination is crucial. The goal is to make these features more discriminative by increasing the inter-dispersion among them, reducing domain biases. This results in more neutral features for unseen target domains. Therefore, using our greedy ensemble (*Index 3 VS. 4*) and enforcing the *uniformity* loss (*Index 5 VS. 6*) leads to positive gains. Appendix F.4 provides visualization using t-SNE (Van der Maaten & Hinton, 2008), and Appendix F.6 reports sensitivity on $\lambda$.

**Domain-aware adaptation and fusion (DAF).** The text and image features $\tilde{\mathbf{I}}(x^{in})$ and $\tilde{\mathbf{T}}^{gre}$ are not inherently tailored to a specific domain. By integrating a domain-aware fusion model, both features are adapted to that domain, facilitating the fusion of text and image modalities within that domain. Hence, *Index 8* demonstrates substantial improvement over domain-agnostic predictions (*Index 6*).

**Training schemes.** ERM samples batches uniformly without considering domain labels, misaligning it with the protocol of adapting to a specific domain using limited unlabeled data. In contrast, our domain-centric learning to adapt optimizes at the domain level, treating each iteration as a task of FSTT-DA. Thus, a domain-centric approach yields further improvements (*Index 7 VS. 8*).

**Effect on number of transformer layers of CPNet.** Fig. 4 illustrates the effect of the number of transformer layers in CPNet. Different downstream datasets require varying learning capacities depending on their complexity. For instance, while DomainNet is more stable, the more challenging remote sensing scenario in FMoW requires additional learnable blocks to achieve reasonable performance(Wang et al., 2022). Nevertheless, even with just 1 layer in CPNet, substantial gains have been observed over zero-shot performance, thanks to the complement of dataset-specific knowledge.

**Analysis on domain prompt DP.** Our domain prompt **DP** aims to adapt the domain-agnostic features $\tilde{\mathbf{I}}(x^{in})$ and $\tilde{\mathbf{T}}^{gre}$ towards a particular domain. It consists of two components: knowledge queried from **K-V** cache, representing source domain knowledge and current domain knowledge $\tilde{D}$ aggregated from its unlabeled data. We omit some parts of **DP**, as reported in Table 4, equipping with both domain knowledge is essential to better adapt the features towards a domain.

**Analysis on domain information aggregation.** VDPG independently processes all the unlabeled data and then aggregates their domain information via averaging. However, we perform simple reshaping and allow the attention to be performed on every pair of the tokens in that data batch which

Table 4: Ablation on domain prompt.

| Domain prompt | iWildCam | | FMoW | |
|---|---|---|---|---|
| | Acc | F1 | WC Acc | Acc |
| **K-V** cache only | 73.0 | 34.1 | 37.8 | 50.6 |
| $\tilde{D}$ only | 71.3 | 32.1 | 35.4 | 50.4 |
| **DP** | 73.4 | 35.2 | 40.9 | 54.8 |

Table 5: Ablation on domain information aggregation.

| Aggregation | iWildCam | | FMoW | |
|---|---|---|---|---|
| | Acc | F1 | WC Acc | Acc |
| Mean | 73.2 | 34.4 | 37.8 | 52.8 |
| Max | 71.9 | 34.5 | 36.7 | 52.6 |
| Reshape $\rightarrow \tilde{D}$ | 73.4 | 35.2 | 40.9 | 54.8 |

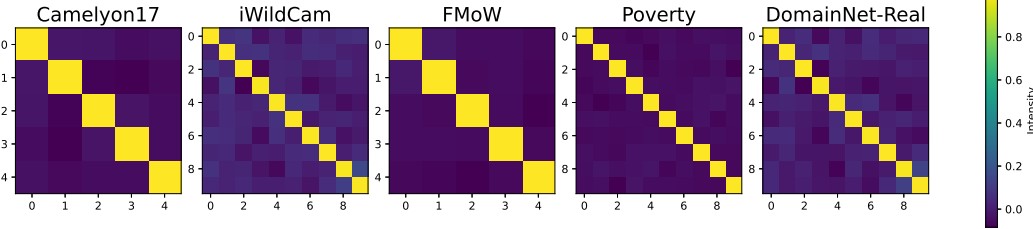

Figure 5: Correlation between every pair of **V**-vectors in **K-V** domain cache.

has shown superiority as reported in Table 5. For Mean and Max, we do not reshape the tensor but take the mean or max over the batch dimension.

**Analysis on K-V domain cache.** $L$ is the size of the domain cache as the number of learnable vectors. Ideally, each can condense the distinct domain specificity from the source domains. Such property is exhibited by computing the correlations between every pair of the **V**-vectors, as illustrated in Fig. 5. Please note, that we did not apply external constraints on the cache (e.g., correlation loss (Chi et al., 2024)). Appendix F.5 reports the sensitivity on the size of the cache.

**Additional analysis on greedy ensemble and text feature uniformity.** Table 6 reports the comparison between the average ensemble (CLIP) and our greedy approach using ViT-B/16 while holding off other components the same. Greater gains on more challenging WILDS benchmarks are observed compared to more structured common objects as in DomainNet. Fig. 6. illustrate that a lower uniformity among text features is potentially beneficial for the final performance.

| Ensemble method | DomainNet (Acc.) | iWildCam (F1) | FMoW (WC Acc) |
|---|---|---|---|
| Ensemble (CLIP) | 60.9 | 33.6 | 40.1 |
| Greedy ensemble | 61.2 | 35.2 | 40.9 |

Table 6: Comparison between average ensemble (CLIP) and our greedy approach using ViT-B/16. The greedy ensemble improves across benchmarks. However, DomainNet contains more structured, common objects. Greater gains on more challenging WILDS benchmarks are observed.

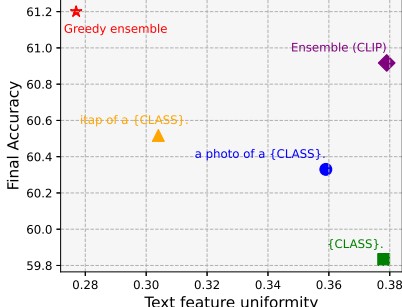

Figure 6: Final accuracy VS. text uniformity on DomainNet with ViT-B/16.

## 6 CONLUSION

In this work, we introduce L2C to address FSTT-DA. L2C adapts a model trained on source domains to unseen target domains using just a few unlabeled data points. Our method builds on the inherent OOD capability of CLIP, complementing it with a parallel network that learns data-specific knowledge at the input space through revert attention. Additionally, we propose a greedy text feature ensemble to effectively integrate data-specific label semantics. To facilitate domain adaptation, we generate a domain prompt that guides the integration of enhanced text and visual features through domain-aware fusion. Our extensive experiments validate L2C's effectiveness, showcasing its superior performance across five large-scale benchmarks in DomainNet and WILDS.

REPRODUCIBILITY STATEMENT

For a fair comparison, we use the VDPG codebase, with data processing following the official WILDS code. The pre-trained CLIP models are directly sourced from the OpenAI CLIP repository. Sample code for the greedy ensemble is provided in Appendix E. Other components, such as CPNet, DAF, text refinement, and K-V cache, utilize standard PyTorch functions. Pre-trained models and the full code will be released upon publication of the paper.

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

# APPENDIX

## A  LIMITATION

Our method enhances the generalized knowledge of robust CLIP, which serve as the primary contributor. However, when the downstream dataset significantly diverges from the pre-training, the load on our CPNet increases, necessitating a larger network. Despite this, our approach focuses on acquiring the excluded knowledge directly from the input space. Consequently, the trade-off between computational demand and performance is more favourable compared to previous methods (VDPG).

## B  ILLUSTRATION FOR FSTT-DA SETTING

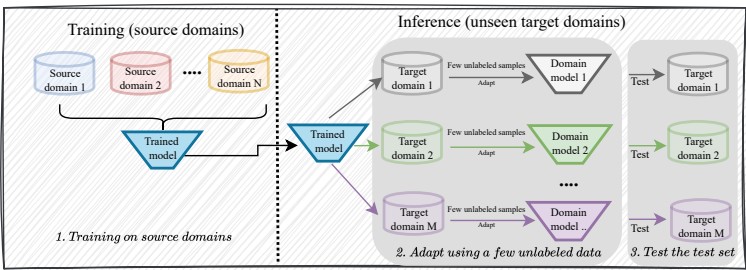

Figure 7: Illustration of FSTT-DA setting. After training on the source domains, the model adapts to each of the target domains using a few unlabeled data samples. Each target domain has a tailored model which will be used to infer all of the data in that domain.

## C  INFERENCE PROCESS (ADAPTING TO A TARGET DOMAIN)

---

**Algorithm 2** Inference: adapting to an unseen target domain

---

**Require:** **I**: CLIP image encoder; $\mathcal{D}_t^m$: an unseen target domain; **CP**: CPNet; **K/V**: K-V domain
   cache; DAF: domain-aware fusion module; $\tilde{\mathbf{T}}^{gre}$: trained text feature;
   1: **// Compute the domain prompt**
   2: $(\mathbf{x}_{\mathcal{S}}) \sim \mathcal{D}_t^m$        ▷ Sample a few unlabeled data samples from the target domain
   3: $\tilde{\mathbf{x}}_{\mathcal{S}}^e \leftarrow \mathbf{x}_{\mathcal{S}}$                 ▷ Form input embeddings
   4: $\tilde{\mathbb{D}} \leftarrow \mathbf{CP}([\mathbb{D}, \tilde{\mathbf{x}}_{\mathcal{S}}^e])$       ▷ Aggregate domain information from unlabeled data
   5: $\mathbf{DP} = [\text{softmax}(\mathbf{K}\tilde{\mathbb{D}}^T) \cdot \mathbf{V}, \tilde{\mathbb{D}}]$     ▷ Compute the domain prompt for domain $\mathcal{D}_t^m$
   6: Discard K-V domain cache
   7: **// Adapting every data in the target domain**
   8: **for** every image $\mathbf{x}_{\mathcal{Q}}$ in $\mathcal{D}_t^m$ **do**
   9:   $\mathbf{x}_{\mathcal{Q}}^{in} \leftarrow \mathbf{x}_{\mathcal{Q}}$               ▷ Form input embeddings
   10:   $\tilde{\mathbf{I}}(\mathbf{x}_{\mathcal{Q}}^{in}) \leftarrow \mathbf{I}(\mathbf{x}_{\mathcal{Q}}^{in}) + \mathbf{CP}^{RT}(\mathbf{x}_{\mathcal{Q}}^{in})$     ▷ Compute complemented visual feature
   11:   $\mathbf{T}_{\mathcal{Q}}^{dm}, \mathbf{I}_{\mathcal{Q}}^{dm} \leftarrow \text{DAF}(\mathbf{DP}, \tilde{\mathbf{I}}(\mathbf{x}_{\mathcal{Q}}^{in}), \tilde{\mathbf{T}}^{gre})$  ▷ Adapt the image $\mathbf{x}_{\mathcal{Q}}$ towards domain $\mathcal{D}_t^m$
   12:   Logits=Cosine_similarity($\mathbf{T}_{\mathcal{Q}}^{dm}, \mathbf{I}_{\mathcal{Q}}^{dm}$)      ▷ Compute predictions
   13: **end for**

---

Algo. 2 shows the process of inference for adapting to a particular unseen target domain. It contains two phases:

**Domain prompt computation.**  For an unseen target domain, we first collect a few unlabeled data samples and compute the domain prompt using the CPNet and the domain cache. Such a step is illustrated in Fig. 8a and L1-5 of Algo. 2. Please note, that after this step, the domain cache can be ignored.

**Adapting the data using the domain prompt.**  Once the domain prompt is computed, it is utilized to adapt all data samples in that domain. This stage follows the process as in L7-12 of Algo. 2. Also as illustrated in Fig. 8b, this step only involved CPNet, CLIP image encoder and proposed DAF.

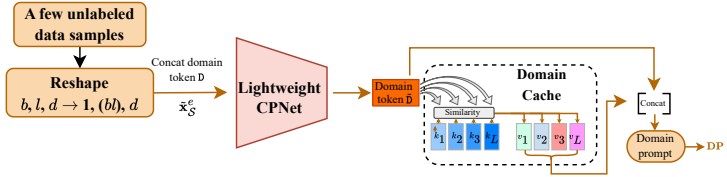

(a) Computing the domain prompt (DP) for the target domain using a few unlabeled data samples.

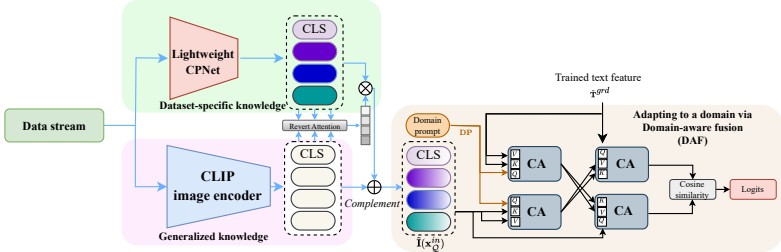

(b) Adapting every data in the target domain.

Figure 8: Inference process for adapting to an unseen domain. Adapting to an unseen target domain involved two phases. The first one is the domain prompt computation using a few unlabeled data sample as in (a). The second phase is to utilize the domain prompt for inferencing all the data in that domain as in (b).

## D COMPUTATIONAL RESOURCES

### D.1 COMPUTATIONAL COMPARISON OVER THE ENTIRE FRAMEWORKS

Table 7: Comparison on speed and memory (batch size of 64).

|  | CPNet(1 layer) | CPNet(3 layer) | CPNet(6 layer) | VDPG |
|---|---|---|---|---|
| Datasets | Camelyon17,PovertyMap | DomainNet,iWildCam | FMoW | All datasets |
| # Learnable params (M) | 13.8 | 27.9 | 49.2 | 32.1 |
| Train memory(MB) | 3872 | 5554 | 8106 | 3672 |
| Train speed(s/batch) | 0.84 | 0.88 | 0.97 | 0.87 |
| Inference memory(MB) | 2270 | 2330 | 2430 | 2798 |
| Inference speed(s/batch) | 0.64 | 0.67 | 0.73 | 0.69 |

Table 7 reports the memory usage and speed during both training and inference. Despite introducing a parallel CPNet, the resource consumption remains comparable to VDPG. While VDPG relies on a heavy guidance module, our framework primarily allocates parameters within CPNet. The increase in memory during training is largely due to tensor reshaping, allowing attention to be applied across the entire batch. However, the domain prompt computation is performed only once per target domain and is gradient-free. As shown in Fig. 8b, the main computation during inference is streamlined—modules such as the text encoder, text feature refinement module, and K-V cache are all eliminated, making our framework highly efficient.

### D.2 EFFICIENCY ON GREEDY TEXT ENSEMBLE.

Greedy Ensemble is executed as a pre-processing step before large-scale training (L3 of Algo. 1). Once the text features for all templates are obtained, the entire text encoder can be discarded, making ensembling highly efficient. For instance, in DomainNet-real with a batch size of 64, the image encoder requires 120K forward passes over 20 epochs, while the text encoder only needs 80 forward passes to compute the text features, resulting in minimal resource usage (<0.01%).

# E  PYTORCH-LIKE SAMPLE CODE FOR GREEDY ENSEMBLE

```python
# P    : Number of text prompt templates
# C    : Number of classes
# d    : feature dimension
# TE   : Tensor, shape=[P, C, d]
#        P sets of text embeddings
# Score: Tensor, shape=[P]
#        corresponding uniformity loss for TE

def uniformity_loss(text_embed, t=2):
    # text_embed: shape=[C, d]
    return torch.pdist(text_embed, p=2).pow(2.0).mul(-t).exp().mean()

def sort_uniformity(TE, Score):
    sort_index = torch.argsort(Score).cpu().numpy()
    return TE[sort_idx]

def ensemble(TE_list):
    return torch.stack(TE_list, dim=0).mean(dim=0)

def greedy_emsenble(TE, Score):
    final_TE = []
    TE_sorted = sort_uniformity(TE, Score)
    # take the text prompt embedding with least uniformoty loss as base
    final_TE.append(TE_sorted[0])

    for i in range(1, P):
        temp_TE = final_TE + TE_sorted[i]
        if uniformity_loss(ensemble(temp_TE))< uniformity_loss(ensemble(final_TE))
            final_TE = temp_TE

    return ensemble(final_TE)
```

# F  ADDITIONAL EXPERIMENTS

## F.1  COMPARISON WITH PROMPT-BASED METHODS AND SIDE BRANCH-BASED METHODS

We further provide a comparison with prompt-based methods: CoOp (Zhou et al., 2022b), Co-CoOp (Zhou et al., 2022a) and side branch-based DTL (Fu et al., 2024). Since these methods are non-adaptive, we applied test-time optimization as in TPT to minimize the entropy (Shu et al., 2022).

Table 8 outlines the architectural differences. TPT relies on gradient updates, making gradient flow crucial and thus limited to white-box settings. In contrast, VDPG and our proposed L2C generate domain-specific prompts for adaptation, enabling them to operate in the more challenging black-box setting. Despite these constraints, L2C still surpasses all other methods, as shown in Table 9.

Table 8: Architectural comparison.

| Method | Black/white box | Gradient at test-time |
|---|---|---|
| CoOp + TPT | white box | required |
| CoCoOp + TPT | white box | required |
| DTL + TPT | white box | required |
| VDPG | black box | gradient-free |
| L2C (Ours) | black box | gradient-free |

Table 9: Performance comparison.

| Method | Ave Acc. (DomainNet, ViT-B16) |
|---|---|
| CoOp | 58.8 |
| CoOp + TPT | 60.6 |
| CoCoOp | 59.4 |
| CoCoOp + TPT | 60.4 |
| DTL | 57.6 |
| DTL + TPT | 59.2 |
| VDPG | 59.8 |
| L2C (Ours) | 61.2 |

## F.2  ALTERNATIVE CRITERIA FOR TEXT EMBEDDING UNIFORMITY

The criteria we used for text feature inter-dispersion among classes is determined by the uniformity in a hypersphere (i.e., Eq. 2). It can also be determined by measuring the Average Text Feature

Dispersion (ATFD) which calculates the distance of all class embedding to their centroid (Yoon et al., 2024). For a text features $\mathbf{T}(\mathbf{P_C})$ with a prompt template P and class labels $\mathbf{C}$, ATFD is computed as:

$$\text{ATFD} = \frac{1}{|\mathbf{C}|} \sum_{i \in |\mathbf{C}|} \|\mathbf{T}_{\text{centroid}} - \mathbf{T}_i(\mathbf{P_C})\|_2, \quad \text{where} \quad \mathbf{T}_{\text{centroid}} = \frac{1}{|\mathbf{C}|} \sum_{i=1}^{|\mathbf{C}|} \mathbf{T}_i(\mathbf{P_C}), \quad (9)$$

where $\|\cdot\|_2$ measures the L2 distance. Please note, a smaller ATFD indicates the class text features are closer to the centroid, therefore, the class features are more closely clustered. In contrast, a larger ATFD indicates a more dispersed distribution of the text features. Therefore, to integrate ATFD into our greedy ensemble pipeline, we need to sort the features of text in reverse order based on ATFD and select the prompts that can **maximize** ATFD after ensemble. The loss function then becomes $\mathcal{L}_{total} = \mathcal{L}_{clip} - \lambda \cdot \text{ATFD}$.

Table 10 reports selected text prompt templates for DomainNet with both $\mathcal{L}_{uni}$ and ATFD. The selected templates match with each other and the average is also very close to each other. Table 11 reports a performance comparison of using $\mathcal{L}_{uni}$ and ATFD on additional benchmarks, showing close results.

Table 10: Comparison between metrics of using two different criteria for text feature inter-dispersion.

|  | DomainNet (ViT-B/16) | |
| --- | --- | --- |
| Inter-dispersion criteria | $\mathcal{L}_{uni}$ | ATFD |
| Selected prompt templates | a blurry photo of a {}. 
 a embroidered {}. 
 itap of the {}. 
 itap of my {}. 
 itap of a {}. 
 a black and white photo of a {}. | a blurry photo of a {}. 
 a embroidered {}. 
 itap of the {}. 
 itap of my {}. 
 itap of a {}. 
 a black and white photo of a {}. |
| Average accuracy on 6 domains | 61.2 | 61.1 |

Table 11: Additional performance comparison of using two different criteria for text feature inter-dispersion.

| Method | iWildCam | | FMoW | | DomainNet |
| --- | --- | --- | --- | --- | --- |
|  | Acc | Macro F1 | WC Acc | Avg Acc | Acc |
| $\mathcal{L}_{uni}$ | 73.4 | 35.2 | 40.9 | 54.8 | 61.2 |
| ATFD | 73.5 | 35.3 | 40.8 | 54.8 | 61.1 |

## F.3 ADOPT CLIP FOR REGRESSION AS IN POVERTYMAP DATASET

**Regression task on CLIP:** Regression task requires a single number output, therefore, it is equivalent to setting the number of output class as 1 (Chi et al., 2024). Since the PovertyMap aims to estimate the wealth index for a region, therefore, we simply use a sentence prompt as the text input of the CLIP text encoder: `A satellite image showing the wealth_index`, yielding a text embedding with output dimension as $\mathbf{T}^{dm} \in \mathbb{R}^{1 \times d}$. With the adapted image feature $\mathbf{I}^{dm} \in \mathbb{R}^{B \times d}$, the logit is obtained by matrix multiplication between them, with shape $\in \mathbb{R}^{B \times 1}$. Therefore, each image has a single output number in our framework. For the regression task, we train the model using MSE loss, and omit the text *uniformity* loss as there is only one "class".

Surprisingly, without any training, the original CLIP model is able to predict zero-shot regression (e.g., the regional wealth index) just like zero-shot classification (as shown in Table 1, the zero-shot regression shows positive correlation on the wealth index). Our method can significantly boost performance by learning the complement knowledge and adapting to those unseen target domains.

## F.4 T-SNE VISUALIZATION OF COMPARISON ON TEXT FEATURES

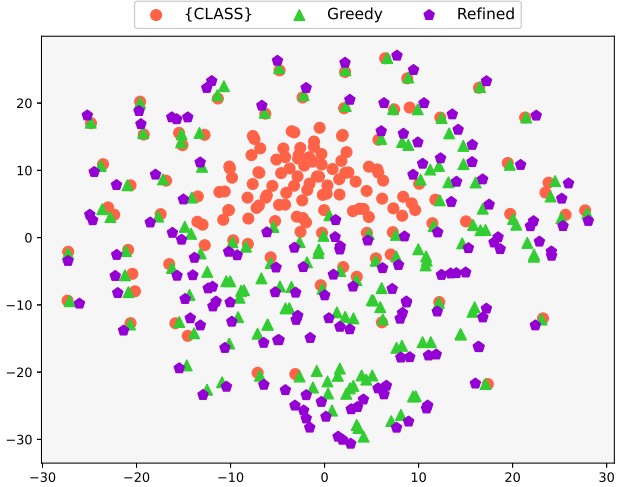

Figure 9: t-SNE (Van der Maaten & Hinton, 2008) visualization of comparison on text features with prompt [CLASS], greedy ensemble and refined output (ViT-B/16 on DomainNet).

Fig. 9 shows the visualization of the text features for different text prompting methods and also our simple yet effective refinement output. It clearly shows that the features with prompt [CLASS] have more clustered features which are less discriminative. Our greedy ensemble greatly increases the distance among the class features. With simple refinement using $M_c, M_d$ and the uniformity loss, the features are further separated.

Table 12: Sensitivity on domain cache size.

| | iWildCam | | FMoW | |
|---|---|---|---|---|
| $L$ size | Acc | F1 | WC Acc | Acc |
| 1 | 69.2 | 32.8 | 33.2 | 50.6 |
| 5 | 73.2 | 35.0 | 40.9 | 54.8 |
| 10 | 73.4 | 35.2 | 40.7 | 54.5 |

Table 13: Sensitivity on loss balancing weight $\lambda$.

| | iWildCam | | FMoW | |
|---|---|---|---|---|
| $\lambda$ | Acc | F1 | WC Acc | Acc |
| 0.01 | 73.2 | 35.1 | 40.8 | 54.7 |
| 0.1 | 73.4 | 35.2 | 40.9 | 54.8 |
| 1.0 | 73.5 | 34.9 | 40.4 | 54.8 |

### F.5 SENSITIVITY ON DOMAIN CACHE SIZE:

Table 12 reports the performance with different size of domain cache. When size=1, the learning capability is too small. Increasing to 5 or 10 makes it more stable.

### F.6 SENSITIVITY ON LOSS BALANCING WEIGHT $\lambda$:

Table 13 reports the sensitivity on $\lambda$. Our framework is less sensitive to $\lambda$ as it is only applied to the text features at the beginning, with only $M_c$ and $M_d$ to optimize. The effect is the convergence speed, but the ultimate performance is quite stable.

## G ADDITIONAL HYPER-PARAMETERS

We utilize the same configurations of transformer blocks as of ViT-B/16 or ViT-L/14, therefore, the feature size can be matched between the main backbone and the CPNet. The only hyper-parameter we tune is the number of transformer blocks. We show the number of layers for different datasets and the total number of learnable parameters and compare them with FLYP and VDPG when ViT-B/16 is utilized in Table14. The size of the domain cache is reported in Table 15. All the experiments can be conducted with a single NVIDIA V100 GPU. We set the batch size as 64 (12 images for support and 52 images for query set). Each iteration runs 0.4 seconds.

Table 14: Configuration of CPNet and the total number of learnable parameters.

| | L2C (ours) | | | | | FLYP | VDPG |
|---|---|---|---|---|---|---|---|
| | DomainNet | iWildCam | Camelyon17 | FMoW | PovertyMap | - | - |
| # of transformer blocks | 3 | 3 | 1 | 6 | 1 | - | - |
| Learnable parameters | 27.9M | 27.9M | 13.8M | 49.2M | 13.8M | 149M | 32.1M |

Table 15: Size of domain cache $L$.

| | DomainNet | iWildCam | Camelyon17 | FMoW | PovertyMap |
|---|---|---|---|---|---|
| $L$ | 10 | 10 | 5 | 5 | 10 |

## H    DETAILS ON TEXT PROMPTS TEMPLATES:

In this section, we describe the candidate test prompts used to perform the greedy ensemble, the full list of attached in supplementary material **(MS Excel spreadsheet (TextPromptTemplates))**. Table 16 shows the selected text prompts among the candidate text prompt as follows:

**DomainNet & iWildCam:**  We use 80 text prompt templates that used for ImageNet (Deng et al., 2009) from CLIP (Radford et al., 2021) and FLYP (Goyal et al., 2023).

**FMoW:**  We use 14 text prompt templates from FLYP (Goyal et al., 2023). The prompts describe the photos in remote-sensing scenarios.

**Camelyon17:**  We filter some text prompts from the 80 ImageNet text prompts that can be used to describe the medical issue and generate some using ChatGPT. In total, there are 56 text prompts.

Please note that PovertyMap is a regression task, the prompts for this task and the experimental setting for regression are reported in Sec. F.3.

Table 16: Selected text prompts by our proposed greedy ensemble. {} is replaced by class name.

| iWildCam | DomainNet | FMoW | Camelyon17 |
|---|---|---|---|
| a blurry photo of the {}. | a blurry photo of a {}. | aerial photo of a {} in oceania. | a dark photo of the {}. |
| a dark photo of the {}. | a embroidered {}. | {}. | a black and white photo of the {}. |
| a cropped photo of the {}. | itap of the {}. | | a good photo of the {}. |
| a black and white photo of the {}. | itap of my {}. | | |
| a black and white photo of a {}. | itap of a {}. | | |
| a close-up photo of the {}. | a black and white photo of a {}. | | |
| a photo of many {}. | | | |
| itap of my {}. | | | |
| a bright photo of the {}. | | | |
| itap of a {}. | | | |
| a good photo of a {}. | | | |

