# OpenReview forum: "Learning to Adapt Frozen CLIP for Few-Shot Test-Time Domain Adaptation"
_ICLR.cc/2025/Conference — ICLR 2025 Poster_

### Official Review · Reviewer_Q3Mk · 2024-10-18

**Soundness:** 3
**Presentation:** 3
**Contribution:** 3
**Rating:** 8
**Confidence:** 3

**Summary:**

This paper introduces "Learning to Complement" (L2C) for Few-shot Test-Time Domain Adaptation, addressing domain shift with minimal unlabeled data. The method adds a parallel module, CPNet, to learn dataset-specific knowledge and complement CLIP’s frozen features. A greedy text ensemble strategy enhances text feature discrimination, improving adaptation by reducing domain bias. The fused text and visual features are guided by domain-specific prompts, leading to superior performance on five benchmarks, especially with smaller backbones like ViT-B/16.

**Strengths:**

*	The organization of this paper is well-structured, making it easy to read and comprehend.

*	The insight of learning dataset-specific knowledge is convincing and effective, making a significant contribution to the FSTT-DA community.

*	Experimental results are impressive, the proposed methods got SOTA in all proposed benchmarks.

*	Overall, the quality of the paper is commendable. The authors have conducted a thorough ablation study to examine the impact of various components.

**Weaknesses:**

*	In Section 4.2, although the authors highlight the improvement of the text dispersion strategy compared to [a], and demonstrate the effectiveness of the greedy text ensemble strategy for FSTT-DA, the advantages of these improvements over previous methods are not sufficiently discussed.

*	The authors did not provide comparisons on the PACS dataset. Extra experiments on smaller datasets could further demonstrate the flexibility of the proposed method.

*	While the paper makes significant progress compared to VDPG and achieves SOTA performance, the authors do not analyze the challenges of FSTT-DA tasks, nor do they provide a discussion on how these challenges are addressed.

[a] E. Cho et al. Distribution-Aware Prompt Tuning for Vision-Language Models. In ICCV, 2023.

**Questions:**

*	The idea of this work is interesting and effective in FSTT-DA scenario, can the authors provide the source code to facilitate reproduction?

*	In line 202, the authors mention that CPNet is introduced to acquire data-specific visual knowledge. However, throughout the paper, CPNet appears to learn dataset-specific knowledge. Could the authors clarify the difference between data-specific visual knowledge and dataset-specific knowledge?

---

I am open to discussing these points with the authors during the response period. If the concerns and questions are adequately addressed, I will consider raising my score.

---

> ### Author Response · Authors · 2024-11-19
> **Response to Reviewer Q3Mk**
>
> Thank you for taking the time to review our paper and provide insightful feedback! We address your concerns as below:
> # **[Weakness]**
>
> > ***1. ... improvement of the text dispersion strategy compared to [A], the advantages of these improvements over previous methods are not sufficiently discussed ...***
>
> We would like to apologize for any confusion caused, which may have led Reviewer Q3Mk to believe that we are comparing our method directly with [A]. To clarify, we are not comparing with [A], but rather integrating the idea of enhancing text dispersion from [A] into our framework. Please allow us to elaborate further:
>
> In [A], a learnable prompt is inserted at the input of CLIP, aiming to increase the dispersion between text features using a dispersion loss to achieve better visual-textual alignment.
>
> In our case, as reported in Fig. 3, for different domains, the image features exhibit large semantic variations across classes. This indicates that the text features should also be adapted accordingly. Therefore, the refined text feature $\tilde{\textbf{T}}^{gre}$, as defined in Eq. 4, serves as a starting point to be adapted for all target domains. To ensure optimal generalization across target domains, it is crucial that $\tilde{\textbf{T}}^{gre}$ does not contain biases, as it is meant to generalize effectively. Thus, we aim to improve the dispersion of $\tilde{\textbf{T}}^{gre}$, ensuring the class prototypes are uniformly distributed. Our greedy ensemble strategy and refinement with the uniformity loss function $\mathcal{L}_{uni}$ all work towards this goal. In Appendix F.2, we also demonstrate that replacing [A] with ATFD achieves the same performance.
>
> Overall, our goal in improving text dispersion is to reduce the bias of text features, making them more generalized and adaptable to different domains. The ablation studies in Table 3, specifically regarding text refinement (Idx 3), greedy ensemble (Idx 4), and uniformity loss (Idx 6), all demonstrate the effectiveness of enhancing the dispersion.
>
> [A] E. Cho et al. Distribution-Aware Prompt Tuning for Vision-Language Models. In ICCV, 2023.
>
> > ***2. ... comparisons on the PACS dataset ...***
>
> We apologize for the missing comparison on PACS. To address this, we conducted experiments on PACS and compared our method with VDPG as follows:
>
>
> |Backnone| PACS Acc (VDPG)|PACS Acc (ours)|
> |-|-|-|
> |ViT-B|96.5|97.8|
> |ViT-L|97.4|98.8|
>
> We used the same hyperparameters as in DomainNet for consistency. As shown, our L2C outperforms VDPG on both backbones, demonstrating the effectiveness of our approach.
>
> > ***3. ... challenges of FSTT-DA tasks ...***
>
>
> We have included the challenges of FSTT-DA task in the revised paper, please see L36-39
>
> The overall challenge of FSTT-DA can be summarized as:
>
>
> 1. **Limited domain-specific information**:
>    The few-shot nature restricts adaptation to a small number of unlabeled target samples, limiting domain-specific cues and increasing the risk of overfitting to noise or spurious patterns.
>
> 2. **One-time adaptation**:
>    Adaptation must occur in a single round for each target domain, requiring efficient and robust mechanisms to generalize effectively across all target data using minimal resources.
>
> 3. **Strict source-free test-time environment**:
>    With no access to source data or labels during test-time, the model must rely solely on target domain cues, making traditional fine-tuning or domain alignment infeasible.
>
> 4. **Diverse target domains**:
>    The method must handle varying domain complexities and shifts, ensuring robustness and scalability across multiple, unpredictable target environments.
>
> # **[Questions]**
>
> > ***1. ... availability of source code ...***
>
> Our source code and pre-trained models to reproduce the results in Tables 1 and 2 will be made publicly available upon the acceptance of the paper.
>
>
> > ***2. ... difference between data-specific visual knowledge and dataset-specific knowledge ...***
>
> we apologize for the typo, we both mean dataset-specific knowledge. We have corrected it in L202 of the revised paper.
>
>
> We sincerely appreciate your time and effort in reviewing our paper and providing insightful comments to help strengthen our work. We hope our rebuttal addresses your concerns, and we remain open to any further questions or clarifications you may have.

---

> > ### Author Response · Authors · 2024-11-23
> > **Follow-up on Rebuttal Discussion**
> >
> > Dear Reviewer Q3Mk,
> >
> > We greatly appreciate the time and effort you have dedicated to reviewing our paper. We hope our rebuttal has addressed your initial concerns. As the discussion phase is nearing completion, we would appreciate any additional feedback to ensure all your concerns are fully resolved. Your insights are invaluable to us, and we are ready to provide further clarifications as needed.
> >
> > Best regards,
> >
> > Authors of Paper 452

---

> ### Comment · Reviewer_Q3Mk · 2024-11-24
>
> Thank you for the author response. I have carefully checked all the reviewer comments, especially the discussion between the authors and reviewer AGPh. I believe all of my concerns, as well as most of the other reviewers' comments, have been addressed. Therefore, I have decided to increase my score.

---

> > ### Author Response · Authors · 2024-11-24
> > **Thank you for your positive support**
> >
> > Dear Reviewer Q3Mk,
> >
> > We are grateful for the time and effort you dedicated to thoroughly reviewing our paper and the other reviewers' comments. We are pleased to hear that you found our responses satisfactory and addressed all concerns, including those raised by other reviewers. Thank you for increasing your score and further supporting our paper.
> >
> > Best regards,
> >
> > Authors of Paper 452

---

### Official Review · Reviewer_pBcp · 2024-10-27

**Soundness:** 3
**Presentation:** 2
**Contribution:** 2
**Rating:** 5
**Confidence:** 4

**Summary:**

This paper presents a new test-time CLIP adaptation method that learns domain-specific knowledge complementary to CLIP’s general knowledge. A parallel CPNet is introduced to achieve this by revert attention from CLIP image encoder. A domain prompt is proposed for domain-aware fusion between text and visual features. The experimental results demonstrate the effectiveness of the proposed method on several benchmarks.

**Strengths:**

1. The experimental results show the improvement over state-of-the-art methods.
2. This paper is well written and follows a good structure.
3. The supplementary material is extensive, offering a lot of supplements and support to the main text.

**Weaknesses:**

1. The motivation behind the proposed method is not clearly explained.  As shown in Figure 1, The method is organized in two branches, fusing domain prompt and complementing dataset-specific knowledge. However, these two branches seem to have very similar functions,  i.e.,  incorporating domain-specific knowledge, which makes the whole pipeline seem redundant.
2. As shown in experiments, it seems that CPNet plays an important role. However, it is not clear that whether the improvement is due to the incorporation of domain knowledge or the addition of extra network parameters.
3. Some relevant prompt-based works exploring domain knowledge for mitigating domain shift is not discussed, e.g., HybridPrompt[1], Pro[2].

    [1] Wu et al., HybridPrompt: Domain-Aware Prompting for Cross-Domain Few-Shot Learning

    [2] Ma et al., ProD: Prompting-to-disentangle Domain Knowledge for Cross-domain Few-shot Image Classification

**Questions:**

1. The introduced modules to refine text prompt, i.e., M_c and M_d are not ablatively evaluated. Can you provide ablation study about this?
2. In Table 3, the performance declines across most of indicators when domain-aware fusion is applied. Why would learning in a domain-aware manner lead to weaker performance?
3. What is the impact of unlabeled samples when learning the target domain prompt? It is advisable to conduct sensitivity analysis of number of samples.

---

> ### Author Response · Authors · 2024-11-19
> **Response to Reviewer pBcp (1/2)**
>
> Thank you for taking the time to review our paper and provide insightful feedback! We address your concerns as below:
>
> # **[Weakness]**
> > ***1. The motivation behind the proposed method is not clearly explained... fusing domain prompt and complementing dataset-specific knowledge seem similar ...***
>
> Thank you for pointing this out. Please allow us to elaborate further. As shown in Fig. 1a, VDPG primarily operates in the feature space of CLIP. While CLIP serves as a generalized knowledge base, its ability to extract domain-specific knowledge is inherently limited.
>
> This limitation motivates us to attach CPNet in parallel with CLIP to directly learn from the input space. In the task of FSTT-DA, there are two stages when performing inference on target domains: 1) adapt using a few unlabeled data, 2) test all the data in that target domain. Therefore, follow these two stages, we expect our CPNet to learn two types of knowledge with two paths:
>
> 1. **Domain-specifc Prompt Computation** (Fig. 8a, Appendix C):
>    This path computes the domain-specific prompt using only a few unlabeled data samples from the target domain, by reshaping the unlabeled data sampling and aggregating domain information to a domain token $(\texttt{D})$ and interact with K-V domain cache, which stores the domain knowledge from source domains. Please note, the Domain-specifc Prompt is **shared for all samples in that domain**.
>
> 2. **Complementing CLIP Features** (Fig. 8b):
>    This path complements CLIP’s generalized features by adding dataset-specific knowledge (e.g., semantics) during inference. This complementing knowledge is instance-based, meaning **every data sample will have its own complement**. The fused features are then guided by the domain-specific prompt computed in the previous step.
>
>
> Intuitively, we could design two separate CPNets, one for each path. However, through experimentation, we found that utilizing a single CPNet for both paths works effectively across all benchmarks. This shared design strikes a balance between computational efficiency and performance.
>
>
>
> > ***2. Not clear that whether the improvement is due to the incorporation of domain knowledge or the addition of extra network parameters ...***
>
>
> Thank you for pointing out this critical concern. We sincerely apologize if the original Fig. 1a gave the impression that our method introduces more parameters compared to VDPG. In fact, VDPG includes heavy learnable modules, whereas our CPNet is designed to be lightweight. To clarify this, we have updated Fig. 1a, Fig. 2, and Fig. 8 in the revised manuscript.
>
> To further address this concern, we direct Reviewer pBcp to Idx 6-7 of Table 3. The Domain-Aware Fusion (DAF) module aims to adapt both complemented visual and textual features using the computed domain-specific prompt. However, as seen in Idx 6 and Idx 7, naively integrating DAF using the ERM training strategy can degrade performance on some metrics. This is because ERM training is unable to effectively extract domain-specific information. Thus, simply introducing additional learnable parameters does not guarantee performance improvement.
>
> On the other hand, the domain-centric training strategy (Idx 8) optimizes at the domain level, treating each iteration as a task of FSTT-DA. This approach aligns the training and evaluation protocols, which is beneficial for computing domain-specific information. Consequently, DAF also benefits from this domain-level optimization.
>
> Furthermore, we would like to direct Reviewer pBcp to Table 7 in Appendix D.1, which compares computational resources between our method and VDPG. For clarity, we also summarize some of the numbers here:
>
> |**Model**|**CPNet (1 layer)**|**CPNet (3 layers)**|**CPNet (6 layers)**|**VDPG**|
> |-|-|-|-|-|
> |**Datasets**|Camelyon17, PovertyMap|DomainNet, iWildCam|FMoW|All datasets|
> |**# Learnable Parameters (M)**|13.8|27.9|49.2|32.1|
>
> As reported, for most benchmarks, our method introduces fewer parameters compared to VDPG. Due to our direct learning from the input space, our method strikes a better balance between learnable parameters and performance. This indicates that the improvement is primarily due to the incorporation of domain-specific knowledge rather than merely adding extra parameters.

---

> ### Author Response · Authors · 2024-11-19
> **Response to Reviewer pBcp (2/2)**
>
> > ***3. Discuss of relevant literature: HybridPrompt and ProD ...***
>
>
> Thank you for pointing out these related works to help make our paper more complete. We have revised the paper to include discussions on HybridPrompt and ProD in L124–127.
>
> Overall, both HybridPrompt and ProD aim to compute effective domain prompts. However, HybridPrompt and ProD address the problem of CD-FSL, which consists of many N-way-K-shot tasks with labeled support data. In contrast, our method focuses on FSTT-DA, which tackles the challenge of adapting models at test time using only a few unlabeled examples from the target domain.
>
> Additionally, HybridPrompt and ProD insert prompts into the middle of the backbone network and require gradient flow for optimization. In contrast, our method operates in a black-box setting, which does not require access to the weights of the CLIP model. This makes our approach more practical and flexible for real-world applications.
>
>
>
> # **[Questions]**
>
> > ***1. Ablation on M_c and M_d ...***
>
> We follow the experimental setup described in Table 3 and insert the ablation study for \( M_c \) and \( M_d \) between Idx 2 and Idx 3.
>
> |**Components**|**Info Acc**|**iWildCam (ACC)**|**iWildCam (F1)**|**FMoW (WC Acc)**|**FMoW (Acc)**|
> |-|-|-|-|-|-|
> |**Idx 2 (without Text ref.)**|49.6 |65.8 |22.8 |28.3 |49.0|
> |**+M_c only**|49.9 |67.1 |23.5 |30.7 |50.1|
> |**+M_d only**|50.1 |67.9 |23.7 |29.9 |49.9|
> |**Idx 3 (M_c and M_d)**|50.3 |68.7| 24.0| 31.4| 50.9|
>
> As shown, introducing \( M_c \) and \( M_d \) individually improves performance, with further improvements observed when both are integrated.
>
>
> > ***2. In Table 3, the performance declines across most of indicators when domain-aware fusion is applied. Why would learning in a domain-aware manner lead to weaker performance?***
>
> Please refer to the answer in Weakness #2.
>
>
> > ***3. impact of unlabeled samples when learning the target domain prompt?***
>
> To ensure a fair and faithful comparison, we followed the inference setting used in MetaDMoE and VDPG, utilizing 16 images per domain. Additionally, we provide a sensitivity analysis on the number of unlabeled data samples for the FMoW dataset:
>
>
> ||1 img|2 imgs|4 imgs|8 imgs|16 imgs|32 imgs|64 imgs|
> |-|-|-|-|-|-|-|-|
> |WC Acc on FMoW|39.8|40.0|40.2|40.6|40.9|41.0|41.0|
>
> From the results above, we observe that performance improves as more unlabeled data is used to extract the domain prompt. The performance becomes less sensitive after 8 images and stabilizes when 32 images are used.
>
>
> We sincerely appreciate your time and effort in reviewing our paper and providing insightful comments to help strengthen our work. We hope our rebuttal addresses your concerns, and we remain open to any further questions or clarifications you may have.

---

> > ### Author Response · Authors · 2024-11-23
> > **Follow-up on Rebuttal Discussion**
> >
> > Dear Reviewer pBcp,
> >
> > We greatly appreciate the time and effort you have dedicated to reviewing our paper. We hope our rebuttal has addressed your initial concerns. As the discussion phase is nearing completion, we would appreciate any additional feedback to ensure all your concerns are fully resolved. Your insights are invaluable to us, and we are ready to provide further clarifications as needed.
> >
> > Best regards,
> >
> > Authors of Paper 452

---

> ### Comment · Reviewer_pBcp · 2024-11-24
> **Response to authors**
>
> The author's response addresses most of my concerns. I decide to maintain my original rating.

---

> > ### Author Response · Authors · 2024-11-24
> > **Glad that most concerns are resolved**
> >
> > Dear Reviewer pBcp,
> >
> > Thank you for acknowledging our efforts to address your concerns in our rebuttal. We are grateful for your thoughtful evaluation and feedback. Given that our rebuttal has addressed most of your concerns, we kindly ask if you would consider revising your rating slightly upwards, as it is currently just below the acceptance threshold.
> >
> > We appreciate your consideration and are happy to provide any further information you might need.
> >
> > Best regards,
> >
> > Authors of Paper 452

---

### Official Review · Reviewer_AGPh · 2024-10-30

**Soundness:** 3
**Presentation:** 3
**Contribution:** 2
**Rating:** 6
**Confidence:** 3

**Summary:**

To address the Few-shot Test-Time Domain Adaptation task, this paper proposes a new method named L2C.

Compared to previous work, L2C adds an input-space learning branch to the frozen CLIP model, overcoming limitations associated with relying solely on CLIP’s feature space. It introduces a greedy text ensemble and domain-aware fusion, enhancing the model's ability to capture dataset-specific semantics.

Extensive experiments on five large-scale benchmarks demonstrate performance improvements, particularly with less robust backbones like ViT-B/16.

**Strengths:**

1. The research task Few-shot Test-Time Domain Adaptation seems practical and meaningful.

2. With less robust backbones like ViT-B/16, the proposed method L2C shows significant performance improvement.

**Weaknesses:**

I have some questions about this paper that need further discussion. Please see them below.

If the authors can address my concerns, I am willing to raise my score.

**Questions:**

I want to disccuss with authors:
## Performance
1. As the author mentioned, the proposed L2C shows better performance with the CLIP ViT-B model, but the performance improvement on the CLIP ViT-L model is limited. Therefore, I would like to know if the proposed L2C method is mainly effective for smaller or "not well-trained" models, and whether L2C might become less effective as models and data scale up.

## Setting
To be honest, Few-shot Test-Time Adaptation (FSTTA) is a new field, so I may have some questions about the settings that I'd like to discuss with the authors.

1. In the traditional TTA setting, once the model is trained on the source domain, most existing works, like TENT [1], make minimal or no changes to the pretrained model, focusing instead on designing methods for the inference stage. However, the L2C method modifies the training process on the source domain (Figure 2). I'm not certain if this approach is permitted.

2. The figure7 shows that the FSTT-DA setting has several target domain, does the FSTTA will occur the catastrophic forgetting like CTTA setting? (CoTTA [2])

3. This method uses prompt guidance and includes multiple domains, which reminds me of PODA [3] and ULDA [4]. I hope the authors can discuss or compare these methods in the paper to enhance its completeness.

## Method
1. As shown in Figure 1, can I understand that, compared to VDPG, L2C introduces a new trainable network to learn more domain-specific knowledge? If so, is the performance comparison fair, given that L2C has more trainable parameters and a larger overall network?

[1] Tent: Fully Test-time Adaptation by Entropy Minimization

[2] Continual Test-Time Domain Adaptation

[3] PØDA: Prompt-driven Zero-shot Domain Adaptation

[4] Unified Language-driven Zero-shot Domain Adaptation

---

> ### Author Response · Authors · 2024-11-19
> **Response to Reviewer AGPh (1/2)**
>
> Thank you for taking the time to review our paper and provide insightful feedback! We address your concerns as below:
>
> # **[Performance]**
>
> > ***ViT-B improves more compared to ViT-L, if the proposed L2C method is mainly effective for smaller or "not well-trained" models ...***
>
>
> We would like to clarify that while L2C demonstrates notable improvements with the smaller ViT-B model, it also provides significant performance gains with larger models, such as ViT-L, particularly on certain datasets like DomainNet. We reuse the evaluation in Table 2 regarding the improvement over VDPG on DomainNet:
>
> |Backnone| Acc averaged on 6 domains (VDPG)|Acc averaged on 6 domains (ours)|
> |-|-|-|
> |ViT-B|59.8|61.2 (+1.4)|
> |ViT-L|65.2|67.4 (+2.2)|
>
> Larger models like ViT-L inherently capture more generalized and domain-agnostic features due to their size and pretraining. As a result, the relative improvement from L2C may appear smaller in some benchmarks because these models already perform well. However, this does not diminish the importance of L2C, as it still enhances performance meaningfully by addressing dataset-specific challenges.
>
> L2C is designed to adapt across a range of model sizes by complementing frozen features with dataset-specific input-level learning and refined text features. As models and data scale up, we expect L2C to remain beneficial, particularly in scenarios involving diverse or challenging domain shifts. This is supported by our results on DomainNet, where even larger models like ViT-L see significant improvements.
>
> # **[Setting]**
>
> > ***1. Is the modification of training process on the source domain allowed? ...***
>
>
> Thank you for bringing this up. We would like to first clarify that our experimental setup strictly follows prior works, such as VDPG and MABN, where training on the source domain is allowed.
>
> In the field of conventional test-time adaptation (TTA), there are two main research directions with distinct setups:
>
> 1. **Test-time Training (TTT):**
>    Customized model training [A,B] (e.g., adding auxiliary tasks or modifying network structures) is performed offline using source data, and the trained model then adapts to test data.
>
> 2. **Fully TTA:**
>    Training on source data is not allowed, and an off-the-shelf pre-trained model is directly used to adapt during inference (e.g., TENT[C], DELTA[D]).
>
>
> **Differentiation between FSTT-DA and conventional TTA:**
> During inference, conventional TTA is instance-based, meaning adapting to **every data sample followed by inference**. However, FSTT-DA is particularly challenging because adaptation is restricted to only a few unlabeled data points per domain. Therefore, the adaptation is only performed **once per target domain**.
>
>
>
> [A] Test-time training with self-supervision for generalization under distribution shifts. Sun et al, ICML2020.
>
> [B] Efficient Test-Time Adaptation for Super-Resolution with Second-Order Degradation and Reconstruction. Deng et al, NeurIPS2023
>
> [C] Tent: Fully testtime adaptation by entropy minimization. Wang et al, ICLR2021
>
> [D] DELTA: degradation-free fully test-time adaptation. Zhao el ta, ICLR2023
>
>
> > ***2. Does the FS-TTDA will occur the catastrophic forgetting like CTTA setting ...***
>
> We apologize for the confusion regarding Figure 7. We would like to clarify that there is no continual evaluation across target domains in FSTT-DA. For each target domain, adaptation starts from the same model trained on the source domain, and the adapted model is only used for that specific target domain. To provide a physical example, as in iWildCam, each camera represents a domain. After training, when we adapt to camera #1, the adapted model will only be used to evaluate data from camera #1.
>
> To further clarify, denoting the model trained on the source domain as $\mathcal{M}$ the evaluation protocol for FSTT-DA is as follows:
>
> **For target domain #1:**
> Adapt $\mathcal{M}$ to create $\mathcal{M}_1$, and use $\mathcal{M}_1$ to evaluate data in target domain #1.
>
> **For target domain #2:**
> Adapt $\mathcal{M}$ to create $\mathcal{M}_2$, and use $\mathcal{M}_2$ to evaluate data in target domain #2.
>
> ... and so on for the remaining target domains.
>
> Thus, if there are $N$ target domains, there will be $N$ distinct adapted models.
>
> We have revised Figure 7 in the updated PDF to reflect this for better understanding. Thank you for pointing this out.

---

> > ### Comment · Reviewer_AGPh · 2024-11-22
> > **A follow up question about the setting**
> >
> > Thank you to the authors for your thorough answers, which seem to address most of my questions.
> >
> > However, I am still confused about the setting of FSTT-DA. In Figure 7, the statement "After training on the source domains, the model adapts to each of the target domains using a few unlabeled data samples" makes me wonder if the setting involves three stages:
> >
> > - First stage: Training the model on the source domains.
> >  - Second stage: Using a few unlabeled data samples to adapt the model, during which the model's parameters are updated.
> > - Last stage: Testing the adapted model on the test set, with the parameters frozen during this phase.

---

> > > ### Author Response · Authors · 2024-11-22
> > > **Response to Reviewer AGPh regarding the follow-up question**
> > >
> > > Dear Reviewer AGPh,
> > >
> > > We sincerely appreciate the time and effort you have dedicated to reviewing our work and providing thoughtful feedback. We are pleased that most of your concerns have been addressed.
> > >
> > > Regarding your follow-up question about the setting:
> > >
> > > You are absolutely correct about the three stages of FSTT-DA. However, we would like to kindly clarify that in the second stage (adapting to each target domain), whether the model parameters are updated depends on the specific method. For instance, in MetaDMoE and MABN, the model parameters are updated, whereas VDPG and our proposed method employ gradient-free adaptation without updating the model parameters. In the third stage, all methods use frozen parameters for testing on the test set.
> > >
> > > We have updated Fig. 7, and hope it provides further clatification.
> > >
> > > We hope this addresses all your concerns. Should you have any remaining questions or require further clarifications, we would be more than happy to assist.
> > >
> > > Best regards,
> > >
> > > Authors of Paper 452

---

> > > > ### Comment · Reviewer_AGPh · 2024-11-22
> > > > **Question about the images selection**
> > > >
> > > > Thank you to the authors for the reply, and I believe I now understand this setting.
> > > >
> > > > In my opinion, this setting is an extension of the traditional few-shot setting. The key difference is that in FSTTA, the adaptation stage occurs during test time, and each sample can only be used once. This is distinct from TTA, where the model is updated during the testing phase without introducing a separate adaptation stage.
> > > >
> > > >
> > > > Hence, I have an additional question. We know that in few-shot tasks, the specific samples selected for adaptation can significantly impact model performance. For FSTTA, since each image can only be seen once, this impact could be further magnified. Therefore, I would like to know: How are the images used for adapting the model selected in FSTTA?

---

> > > > > ### Author Response · Authors · 2024-11-22
> > > > > **Response to Reviewer AGPh regarding the images selection**
> > > > >
> > > > > Dear Reviewer AGPh,
> > > > >
> > > > > We are delighted to hear that our previous clarification was helpful in providing a better understanding of the FSTT-DA setting.
> > > > >
> > > > > Regarding your question about image selection for each domain:
> > > > >
> > > > > We strictly adhered to the evaluation protocol outlined in MetaDMoE, MABN, and VDPG, utilizing the dataloaders provided in the WILDS codebase (Koh et al., 2021) for all benchmarks. During testing, the dataloader first samples one target domain and then sequentially loads the images from that domain. We use the first 16 images for adaptation, ensuring consistency across experiments by using the same random seeds. This guarantees that the selected 16 images are identical to those used in MetaDMoE, MABN, and VDPG, facilitating a fair comparison.
> > > > >
> > > > > To provide a real-world example, consider the iWildCam benchmark, where each camera is treated as a domain. When a camera is installed, there is typically a calibration phase during which the camera captures a few initial images. These images can be used for adaptation. Once the calibration is complete, our model can be directly deployed on that camera.
> > > > >
> > > > > We hope this explanation clarifies your concerns. Should you have any further questions or require additional details, we would be more than happy to assist.
> > > > >
> > > > > Best regards,
> > > > >
> > > > > Authors of Paper 452

---

> > > > > > ### Comment · Reviewer_AGPh · 2024-11-22
> > > > > > **Score raising and looking forward to further analysis**
> > > > > >
> > > > > > Thank you to the authors for providing clarifications. Since most of my concerns have been addressed, I am raising my score to 5.
> > > > > >
> > > > > > Besides, in my opinion, due to the FSTT-DA setting, which requires the model to see each image only once during the adaptation stage, the number and diversity of images are likely to have a more significant impact compared to the traditional few-shot setting. If the authors could provide additional explanations or experimental results to analysis this aspect, I would consider further raising my score.

---

> ### Author Response · Authors · 2024-11-19
> **Response to Reviewer AGPh (2/2)**
>
> > ***3. Discussion with PODA and ULDA ...***
>
> Thank you for pointing out the related works and enhancing our completeness. We have included the discussion of PØDA and ULDA in the revised PDF (L111–L114).
>
> The main difference is that ULDA and PØDA focus on leveraging textual descriptions for adaptation, making them highly flexible for scenarios where collecting domain-specific descriptions is feasible.
>
> In contrast, FSTT-DA adapts based on a small number of unlabeled target image. This is practical for real-world applications like medical imaging or wildlife monitoring, where obtaining descriptions or labels may not always be possible. To elaborate, the prompts for ULDA and PØDA are based on natural language, whereas our domain prompts require images as input. Furthermore, ULDA and PØDA focus on segmentation tasks, while FSTT-DA primarily addresses classification tasks.
>
> While ULDA and PØDA excel in generalization across a wide range of domains, FSTT-DA offers a specialized approach for scenarios where few-shot, target-specific adaptation is essential. Each method has distinct strengths, and their integration could open new avenues for robust domain adaptation research.
>
> # **[Method]**
> > ***Is the performance comparison fair, given that L2C has more trainable parameters and a larger overall network? ...***
>
>
> Thank you for pointing this out. We sincerely apologize if the original Fig. 1a gave the impression that our method introduces more parameters compared to VDPG. In fact, VDPG includes heavy learnable modules, whereas our CPNet is designed to be lightweight. To clarify this, we have updated Fig. 1a, Fig. 2, and Fig. 8 in the revised manuscript.
>
> We would like to refer Reviewer AGPh to Table 7 in Appendix D.1, which compares trainable parameters, speed, and memory. To clarify further, we provide the following summary:
>
> |**Model**|**CPNet (1 layer)**|**CPNet (3 layers)**|**CPNet (6 layers)**| **VDPG**|
> |-|-|-|-|-|
> |**Datasets**|Camelyon17, PovertyMap|DomainNet, iWildCam|FMoW              |All datasets|
> |**# Learnable Parameters (M)**|13.8| 27.9| 49.2|32.1|
>
> As shown, for most benchmarks (except FMoW), our method introduces fewer trainable parameters compared to VDPG. It is important to note that the table above reflects the number of parameters used in our **best-performing models**.
>
> To further elaborate, we also record the **minimum number of layers** required for our method to surpass or perform comparably to VDPG:
>
> |**Dataset**|**Minimum Number of Layers to Surpass VDPG**|**Corresponding Parameters (M)**|
> |-|-|-|
> |Camelyon17|1| 13.8M|
> |PovertyMap|1| 13.8M|
> |Infograph|1| 13.8M |
> |iWildCam|2| 20.9M|
> |FMoW|5|42.1M|
>
> For most benchmarks, our method requires fewer trainable parameters than VDPG. The key distinction lies in the design philosophies of the two methods:
> - VDPG solely relies on the generalized features of CLIP and therefore requires **heavy modules (e.g., generator and guidance) to learn meaningful domain information**.
> - In contrast, our method directly learns from the input space, which is more effective for capturing domain-specific knowledge. This allows our approach to achieve a **better trade-off between resource usage and performance**.
>
> We sincerely appreciate your time and effort in reviewing our paper and providing insightful comments to help strengthen our work. We hope our rebuttal addresses your concerns, and we remain open to any further questions or clarifications you may have.

---

> ### Author Response · Authors · 2024-11-22
> **Follow-up response regarding impact of image selection**
>
> Dear Reviewer AGPh,
>
> We sincerely appreciate your thoughtful feedback and are delighted that most of your concerns have been addressed. We are especially grateful that you have raised the score for our submission.
>
> To provide further insights regarding the impact of different images at the adaptation stage, we conducted additional experiments followed by a detailed analysis.
>
> We evaluated the FMoW dataset using CLIP-ViT/B16 with different batches of 16 images during the adaptation stage. The results are reported as worst-case accuracy (WC Acc) for two models: one trained without domain-centric training (ERM, Index 7 of Ablation Table 3) and another trained with domain-centric training (Index 8 of Ablation Table 3).
>
> |Training Scheme|Trial #1 (same as paper)|Trial #2 |Trial #3|Trial #4 |Mean/Var|
> |-|-|-|-|-|-|
> |Without domain-centric (ERM, Index 7)| 36.0|37.2|37.8|34.7|36.4/1.4|
> |With domain-centric (Index 8)|40.9|41.2|40.4|40.9|40.9/0.08|
>
> As shown in the table, our full method demonstrates greater robustness to variations in image selection compared to the model trained without domain-centric training (ERM). We would like to elaborate further on this observation:
>
> The domain-specific knowledge within a domain should ideally be both instance-agnostic and class-agnostic, meaning that every data sample from the same domain inherently carries similar domain properties. For instance, in the painting domain of DomainNet, data samples should resemble **paintings** rather than **real** natural images. Thus, effectively extracting domain-specific knowledge during the adaptation stage is critical in FSTT-DA.
>
> The ERM training scheme draws data batches uniformly from various source domains. This approach is not optimized for focusing on domain-specific information, resulting in greater sensitivity to the selection of images. In contrast, domain-centric training simulates the evaluation protocol by mimicking the process of adapting to a single domain in episodic tasks. Through numerous simulated FSTT-DA tasks during training, our domain branch becomes effective in extracting domain-specific knowledge. Consequently, our method is less impacted by image selection variations during the adaptation stage.
>
>
> ***Discussion regarding the comparison with conventional few-shot learning (FSL):***
> We would like to first clarify that the label space for FSL changes across few-shot tasks, whereas the label space for FSTT-DA remains the same across domains. FSL typically involves evaluations on N-way K-shot tasks, with a total of NxK labeled samples in the support set and an unlabeled query set. During the learning stage on the support set, the model must learn both class information and, if domain shift is present, domain information. However, class information often varies significantly across data samples. If there are large disparities in class information between the support and query sets, performance will degrade significantly.  Therefore, the impact of the images selected for the support set also depends heavily on how the benchmark is constructed (i.e., the closeness of the support and query sets). This highlights a key difference in focus between FSTT-DA and FSL: FSTT-DA focuses on effectively extracting domain information from unlabeled data, whereas FSL emphasizes rapid adaptation to novel classes by learning appropriate class information.
>
> To demonstrate the impact of varying the number of unlabeled images during the adaptation stage, we conducted experiments using different quantities of images:
>
> | Number of Images | 1 img | 2 imgs | 4 imgs | 8 imgs  | 16 imgs (used in paper) | 32 imgs | 64 imgs |
> |-|-|-|-|-|-|-|-|
> | WC Acc on FMoW (ViT-B/16) | 39.8  | 40.0   | 40.2   | 40.6   | 40.9    | 41.0    | 41.0    |
>
> From these results, we observe that performance improves as more unlabeled data is used to extract the domain prompt. The performance becomes less sensitive after eight images and stabilizes when 32 images are used. This aligns with our expectations, as the domain information becomes more complete with an increasing number of data samples. To sum up, with carefully designed techniques, such as optimized training schemes and domain knowledge extraction modules, the effects of variations in adaptation images can be mitigated, provided the domain knowledge is effectively captured in FSTT-DA.
>
> We sincerely thank you for your time and effort in reviewing our paper and for your thoughtful suggestions, which have made our work stronger. We hope the additional results and explanations address your concerns. We remain open to addressing any further questions you may have.
>
> Best regards,
>
> Authors of Paper 452

---

> > ### Comment · Reviewer_AGPh · 2024-11-23
> > **Response to Authors**
> >
> > Thank you to the authors for providing clarifications. Most of my concerns have been addressed, I am raising my score to 6.

---

> > > ### Author Response · Authors · 2024-11-23
> > > **Thank you**
> > >
> > > Dear Reviewer AGPh,
> > >
> > > Thank you for increasing the score to positive. We appreciated the opportunity to engage in a constructive discussion with you during the rebuttal process, which provided valuable feedback and helped strengthen our paper.
> > >
> > > Best regards,
> > >
> > > Authors of Paper 452

---

### Official Review · Reviewer_aphZ · 2024-11-02

**Soundness:** 3
**Presentation:** 3
**Contribution:** 2
**Rating:** 6
**Confidence:** 4

**Summary:**

This paper introduces a novel approach for few-shot test-time domain adaptation (FSTT-DA) by learning to adapt frozen CLIP models to specific domains using only a few unlabeled examples. The method leverages strong CLIP`s OOD abilities and further learns on the input space to complement CLIP's generalized features with dataset-specific knowledge. Additionally, it enhances text features through a greedy ensemble strategy and proposes a domain-aware fusion approach for adapting both text and visual features toward a target domain.

**Strengths:**

1. The overall writing is good and clear.
2. Promising results are obtained in the studied benchmarks.
3. Comprehensive evaluations are conducted, and the effectiveness of the proposed modules is also shown.

**Weaknesses:**

Novelty is somewhat limited.  I appreciate the proposed method improved the base CDPG obviously and the proposed method (as in Fig.2) also includes many submodules. However, except for the greedy text ensemble strategy, the techniques used in the submodels e.g., domain prompt, and cross-attention-based fusion all look not new.

**Questions:**

1. what are the additional number of parameters and computation costs?
2. In Lines 510-511, the paper mentions that compared to previous work external constraints are not used. I am curious why these losses are not used. Will the results become worse when using them?
3.	How does the similarity between the target domain and the source domain affect the method?
2.	In Fig. 4, for some datasets, the increasing number of layers (parameters) of CPNet can not bring further improvement in accuracy but makes the accuracy lower.  What is the reason behind this?
5.   Open question: why the FSTT-DA task with few-shot unlabeled data is selected as the testbed, what about the few-shot labeled data but also with the cross-domain (CD-FSL) [ref1,2,3] setting?  To me, the unlabeled and small labeled data don't come together quite often.

[ref1]  A Broader Study of Cross-Domain Few-Shot Learning
[ref2] Cross-Domain Few-Shot Classification via Adversarial Task Augmentation
[ref3] StyleAdv: Meta Style Adversarial Training for Cross-Domain Few-Shot Learning

---

> ### Author Response · Authors · 2024-11-19
> **Response to Reviewer aphZ (1/3)**
>
> Thank you for taking the time to review our paper and provide insightful feedback! We address your concerns as below:
>
> # **[Weakness]**
> > ***Novelty is some what limited ... the proposed method improves SoTA VDPG obviously ... greedy text ensemble strategy is novel but some sub-modules look not new ...***
>
>
> We appreciate Reviewer aphZ for recognizing our obvious improvement over VDPG, the SoTA method for FSTT-DA, and acknowledging our novel greedy text ensemble strategy.
>
> We would like to respectfully argue that novelty should be evaluated based on the overall framework rather than individual submodules. While the concept of some components may not be entirely new, our Learning to Complement (L2C) framework introduces a novel integration and re-imagining of these techniques, specifically tailored to the under-explored Few-shot Test-Time Domain Adaptation (FSTT-DA) task. We believe that tailor such techniques for challenging, under-explored learning paradigms to achieve significant improvements constitutes a meaningful contribution.
>
> To support this perspective, we provide examples of impactful works that integrate existing components for new tasks:
> - **CoOp (Zhou et al., 2022b)**: A prompt-based method that applies NLP's prompt tuning[A] directly to the CLIP model.
> - **FLYP (Goyal et al., 2023)**: A fine-tuning baseline that directly uses the CLIP model and training scheme without adding any new network components or objectives for downstream tasks.
> - **Model soups (Wortsman et al., 2022a)**: Simply ensembles model parameters to achieve a better trade-off between in-distribution (ID) and out-of-distribution (OOD) performance.
> - **MABN (Wu et al., 2024)**: Identifies the affine parameters in batch normalization layers to learn domain-specific knowledge, which is a common approach for tackling domain shift and test-time adaptation[B].
>
> We understand the importance of novelty in a submission and would like to clarify the unique contributions of our work:
>
> 1. **Addressing VDPG's Limitations**: As stated in Sec. L52–L70, L160–L193, and Fig. 1(a), VDPG's reliance on frozen CLIP features limits its ability to learn dataset-specific knowledge. In contrast, L2C is designed to address these limitations with a black-boxed architecture.
>
> 2. **A Holistic Framework**: L2C is a unified framework to combine input-level learning, enhanced text adaptation, and domain-aware fusion in a unified manner for FSTT-DA. Each component is carefully integrated to tackle VDPG's limitations and achieve SoTA performance on challenging benchmarks.
>
> 3. **Practical Effectiveness**: The integration of domain-cache-based prompts and domain-aware fusion, alongside CPNet's lightweight design, demonstrates significant improvements. Our ablation studies in Sec. 5.2 clearly show that each individual component contributes to the overall performance gains.
>
>
> [A] The Power of Scale for Parameter-Efficient Prompt Tuning, Lester et al, EMNLP21.
>
> [B] Tent: Fully Test-time Adaptation by Entropy Minimization, Wang et al, ICLR21.

---

> ### Author Response · Authors · 2024-11-19
> **Response to Reviewer aphZ (2/3)**
>
> # **[Questions]**
>
> > ***1. Additional number of parameters and computation costs ...***
>
>
> We sincerely apologize if the original Fig. 1a gave the impression that our method introduces more parameters compared to VDPG. In fact, VDPG includes heavy learnable modules, whereas our CPNet is designed to be lightweight. To clarify this, we have updated Fig. 1a, Fig. 2, and Fig. 8 in the revised manuscript.
>
>
> We refer Reviewer aphZ to Table 7 in Appendix D.1, which provides detailed comparisons of learnable parameters, memory usage, and speed. For clarity, we reuse the table here. Overall, our framework is more effective because it directly learns at the input space, requiring fewer resources on most benchmarks compared to VDPG. In contrast, VDPG relies on heavy training modules constrained by CLIP's generalized knowledge.
>
> |Batch size=64|CPNet(1 layer)|CPNet(3 layer)|CPNet(6 layer)|VDPG|
> |-|-|-|-|-|
> |Datasets|Camelyon17,PovertyMap|DomainNet,iWildCam|FMoW|All datasets|
> |# Learnable parameters(M)|13.8|27.9|49.2|32.1|
> |Train memory(MB)|3872|5554|8106|3672|
> |Train speed(s/batch)|0.84|0.88|0.97|0.87|
> Inference memory(MB)|2270|2330|2430|2798|
> Inference speed(s/batch)|0.64|0.67|0.73|0.69|
>
> Our whole framework can be broken down into the following steps:
>
> 1. **Pre-processing:**
>    Our greedy ensemble is executed as a pre-processing step before large-scale training. Therefore, the text encoder of CLIP can be discarded after this step, as shown in L3 of Algorithm 1. This part consumes minimal resources (<0.01%), as detailed in Appendix D.2.
>
> 2. **Training:**
>    While our framework involves reshaping the unlabeled tensor to compute the domain prompt, which requires additional memory, the training speed is comparable to or slightly faster than VDPG.
>
> 3. **Inference:**
>    Our inference process is illustrated in Appendix C and consists of two parts:
>    - **Domain prompt computation:** This step (Fig. 8a) is only performed once for each domain.
>    - **Adaptation to the domain:** After the prompt is computed, the domain branch can be discarded for subsequent inferences (Fig. 8b).
>
> We emphasize that the inference stage is more important because training is performed only once on the source domains, while the trained model is used for inferring all target domains.
>
> > ***2. External constraints are not used as in VDPG ...***
>
> The correlation loss function in VDPG, as mentioned in L511, aims to enforce that the knowledge learned on the source domain is orthogonal, effectively removing redundant knowledge. However, our empirical results in Fig. 5 already demonstrate the orthogonality property, even without this constraint.
>
> To further analyze its impact, we integrate the correlation loss into our training pipeline and test it on iWildCam and FMoW using ViT-B/16. The results below demonstrate that this loss has minimal impact on our method:
>
>
> |Backnone|iWildCam (Macro F1)|iWildCam (Macro F1)|FMoW (WC Acc)|FMoW (Acc)|
> |-|-|-|-|-|
> |without correlation loss|73.4|35.2|40.9|54.8|
> |with correlation loss|73.5|35.2|40.9|54.7|
>
>
> > ***3. Effect of similarity between source and target domain for our method ...***
>
> The similarity between the target and source domains can influence performance in domain shift method (VDPG, MABN, MIRO, etc) , as higher similarity typically allows for better knowledge transfer. However, our method is specifically designed for robust knowledge transfer among domains.
>
>
>
> 1. **Robustness to Domain Shifts**:
>    Our framework uses CPNet to complement CLIP's generalized features with dataset-specific knowledge, enabling adaptation even for significantly different target domains.
>
> 2. **Domain-Specific Adaptation**:
>    Domain-aware prompts tailor text and image features to the target domain by incorporating both source and target-specific knowledge (Eq. 5).
>
> 3. **Empirical Evidence**:
>    Experiments (e.g., Tables 1 and 2) show our method consistently outperforms prior approaches on benchmarks like iWildCam, FMoW, and DomainNet, demonstrating robustness across diverse domains, including significantly dissimilar ones. Especially, as demonstrated in DomainNet in Table 2, **each of the 6 domains is quite different from others**, our method shows robustness and achieves superior average performance.

---

> ### Author Response · Authors · 2024-11-19
> **Response to Reviewer aphZ (3/3)**
>
> > ***4. In Fig. 4, some datasets may have lower accuracy with the increase in layers of CPNet ...***
>
> We hypothesis that this phenomenon occurs due to the following reasons:
>
> 1. **Overfitting on limited data (as in general ML paradigms):**
>    Increasing the number of layers or parameters can lead to overfitting. This is particularly relevant for the WILDS benchmarks, which are highly imbalanced at both the domain and instance levels. Some domains may contain only a few classes with very limited samples per class. With more complex models, CPNet may learn spurious patterns or noise specific to the target domain samples, negatively impacting generalization.
>
> 2. **Interference with CLIP's generalized knowledge:**
>    CPNet is designed to complement CLIP by learning dataset-specific knowledge while preserving CLIP's generalized features. Adding excessive layers may cause CPNet to dominate the feature fusion process, potentially introducing biases that interfere with the generalization capabilities of the frozen CLIP model.
>
> 3. **Capacity misalignment:**
>    For some datasets, the complexity of the task may not require additional parameters. In these cases, adding more layers increases the model’s capacity without providing a meaningful advantage, leading to optimization challenges and reduced accuracy.
>
>
> > ***5. why the FSTT-DA task with few-shot unlabeled data is selected as the testbed but not the few-shot labeled data as in cross-domain few-shot learning(CD-FSL)***
>
> WILDS and DomainNet are well established benchmarks for FSTT-DA, by the previous works (VDPG, MetaDMoE). We followed their experimental setting for faithful and fair comparison. We ellaborate the difference between FSTT-DA and CD-FSL:
>
> 1. **FSTT-DA:**
> FSTT-DA is more related to domain generalization (DG) where the training is performed on the source domains and test on target domain. On top of DG, FSTT-DA focuses on the test-time capability and introduces **an additional learning phase at test-time** to adatp to each of the target domains using a few unlabeled data. Therefore, DG benchmarks, WILDS and DomainNet, are directly adopted and required no additional labeling for FSTT-DA. As in many applications, such as medical imaging or wildlife monitoring, collecting unlabeled samples from the target domain is feasible, but annotating these samples requires significant effort, expertise, or time. This makes the few-shot unlabeled setting more common than the few-shot labeled scenario in these contexts. On the other hand, FSTT-DA assumes that the labeled space is the same across domains as in DG.
>
> 2. **CD-FSL:**
> CD-FSL is more related to few-shot learning but encouteres domain shift for the few-shot evaluation. Please note, CD-FSL assumes that label space is changing for every N-way-K-shot task.
>
> Overall, the FSTT-DA setting allows for seamless adaptation across a wide range of domains without requiring task-specific annotations. This scalability makes it suitable for dynamic or resource-constrained environments. In the revised PDF, we also discussed the 3 mentioned literature as in L124 to L127.
>
> We sincerely appreciate your time and effort in reviewing our paper and providing insightful comments to help strengthen our work. We hope our rebuttal addresses your concerns, and we remain open to any further questions or clarifications you may have.

---

> > ### Author Response · Authors · 2024-11-23
> > **Follow-up on Rebuttal Discussion**
> >
> > Dear Reviewer aphZ,
> >
> > We greatly appreciate the time and effort you have dedicated to reviewing our paper. We hope our rebuttal has addressed your initial concerns. As the discussion phase is nearing completion, we would appreciate any additional feedback to ensure all your concerns are fully resolved. Your insights are invaluable to us, and we are ready to provide further clarifications as needed.
> >
> > Best regards,
> >
> > Authors of Paper 452

---

> > > ### Comment · Reviewer_aphZ · 2024-11-26
> > > **Respond to the Follow-up on Rebuttal**
> > >
> > > Dear authors,
> > >
> > > Thanks for the detailed responses and clarifying the questions.  Most of my concerns are being addressed, I agree that the proposed overall framework addresses the tackled task with improvements achieved, but I still think the technical part doesn't bring new things to me. Also since my original score is already positive, I will keep the same score.

---

> > > > ### Author Response · Authors · 2024-11-26
> > > > **Thank you for your positive support**
> > > >
> > > > Dear Reviewer aphZ,
> > > >
> > > > Thank you very much for your thoughtful review and for taking the time to carefully review our responses. We are grateful for your positive evaluation of the overall framework and the improvements it offers in addressing the task at hand. We are pleased that our responses have addressed most of your concerns. Your continued support and the positive score are greatly appreciated.
> > > >
> > > > Best regards,
> > > >
> > > > Authors of Paper 452

---

### Meta-Review · Area_Chair_hpGR · 2024-12-15

**Metareview:**

This paper tackles Few-shot Test-Time Domain Adaptation by enabling models to adapt during testing using only a few unlabeled examples, effectively addressing domain shift limitations in models like CLIP. By introducing a side branch to capture dataset-specific knowledge and enhancing feature fusion with domain prompts, it achieves notable performance improvements on challenging benchmarks. While one reviewer initially expressed concerns, the authors have successfully addressed most of these issues, and all other reviewers are positive and satisfied with the revisions and responses.

**Additional Comments On Reviewer Discussion:**

All comments are addressed.

---

### Decision · Program_Chairs · 2025-01-22

Accept (Poster)